# DiffEM: Learning from Corrupted Data with Diffusion Models via Expectation Maximization

## Abstract

Diffusion models have emerged as powerful generative priors for high-dimensional inverse problems, yet learning them when only corrupted or noisy observations are available remains challenging. In this work, we propose a new method for training diffusion models with Expectation-Maximization (EM) from corrupted data. Our proposed method, DiffEM, utilizes conditional diffusion models to reconstruct clean data from observations in the E-step, and then uses the reconstructed data to refine the conditional diffusion model in the M-step. Theoretically, we provide monotonic convergence guarantees for the DiffEM iteration, assuming appropriate statistical conditions. We demonstrate the effectiveness of our approach through experiments on various image reconstruction tasks.

## 1 Introduction

Diffusion models (Song and Ermon, 2019; Ho et al., 2020; Song et al., 2020) have emerged as powerful tools for learning high-dimensional distributions, achieving remarkable success across a broad range of generative tasks. Their effectiveness as learned priors has led to significant advances in solving inverse problems (Kawar et al., 2021; Choi et al., 2021; Saharia et al., 2022), including image inpainting, denoising, and super-resolution. However, in many real-world scenarios, acquiring clean training data remains difficult or costly, and can raise significant concerns, as training on clean data might lead to memorization (Somepalli et al., 2023a; Carlini et al., 2023; Somepalli et al., 2023b; Shah et al., 2025), posing privacy and copyright risks. While data with mild or moderate corruption is often more readily available, particularly in domains like medical imaging (Wang et al., 2016; Zbontar et al., 2018) and compressive sensing, training diffusion models effectively using only corrupted or noisy observations presents substantial technical challenges.

The fundamental difficulty lies in the fact that standard techniques for training diffusion models are designed for settings with access to clean data from the data distribution. When only corrupted or noisy observations are available, these techniques become inapplicable, and training diffusion models effectively reduces to learning a latent variable model from corrupted observations, a problem well-known for its theoretical and practical challenges.

Recent work (Rozet et al., 2024; Bai et al., 2024) has proposed addressing this challenge by applying the Expectation-Maximization (EM) method with diffusion models as priors. However, this approach faces a critical difficulty: in each E-step, the algorithm must sample from the posterior distribution given the corrupted observations, whereas it only has access to the score function of the diffusion prior. To overcome this, these works adopt ad hoc posterior sampling schemes that rely on various approximations of the posterior score function that explicitly incorporate the likelihood function. Such approximation schemes, however, are based on implicit structural assumptions about the true data distribution and the likelihood function, making their approximation errors difficult to quantify.

In this work, we propose a new method that combines diffusion models with the EM framework. Our key insight is that instead of learning a diffusion prior and then performing approximate sampling, we can directly model the posterior distribution using a conditional diffusion model (Saharia et al., 2022; Daras et al., 2024a). The primary advantage of our approach is its independence from specific approximate posterior sampling schemes. Notably, it can handle any likelihood function, as it makes no assumptions about the data distribution and likelihood function beyond requiring that the posterior score function can be expressed by the denoiser network. Furthermore, we provide theoretical

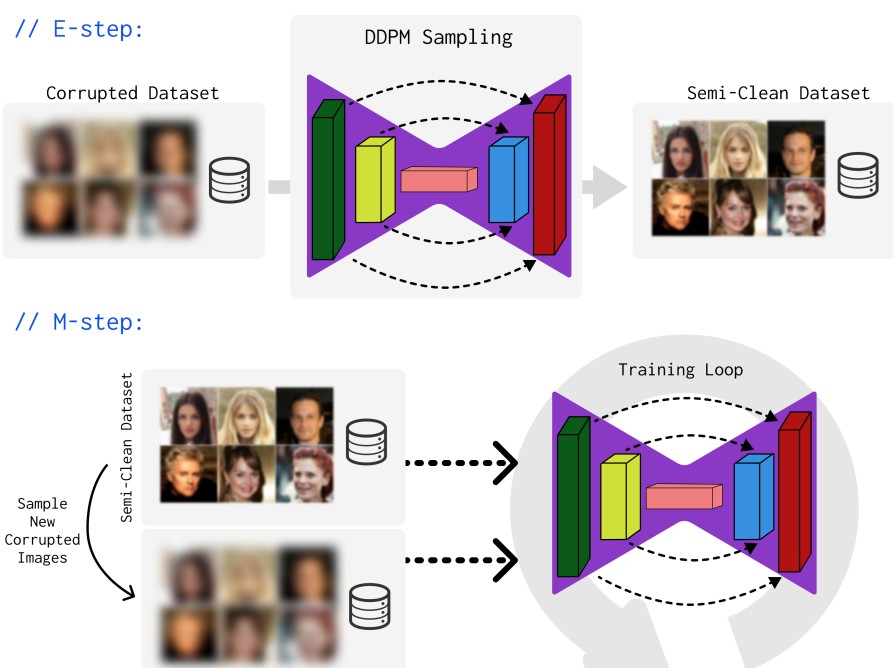

Figure 1: Illustration of how the algorithm runs each EM iteration, on the top there is the Expectation step where the conditional diffusion model generates samples and in the bottom is the Maximization step where the diffusion model is trained using the generated data.

analysis of the proposed EM iteration, demonstrating its convergence under appropriate conditions on the approximation error of the denoiser network. We validate our approach through extensive experiments on both synthetic and real-world datasets with various types of corruption, including low-dimensional manifold learning and unconditional generation on CIFAR-10 and CelebA.

**Related work.** Due to space limitations, we discuss further related work in Appendix A, and provide more detailed discussions of the closest works to ours in the next couple of sections.

## 1.1 PRELIMINARIES

**Problem setup.** Formally, we consider the following setup. The *data distribution* $P_X^\star$ is a distribution over the space $\mathcal{X}$ of latent variables, and the *likelihood* $\mathbf{Q}(\cdot|X)$ maps each point $X \in \mathcal{X}$ to a distribution over the observation space $\mathcal{Y}$. The observation is generated as

$$Y \sim \mathbf{Q}(\cdot|X), \quad \text{where } X \sim P_X^\star, \tag{1}$$

and we denote $P^\star$ to be the joint distribution of $(X, Y)$ and $P_Y^\star$ to be the marginal distribution of $Y$. This formulation encompasses classical inverse problems by specifying $\mathbf{Q}(\cdot|X) = \mathsf{N}\big(\mathcal{A}(X), \sigma_Y^2 \mathbf{I}\big)$, where $\mathcal{A} : \mathcal{X} \to \mathbb{R}^d$ is a known forward operator.

In our setting, the learner only has access to a dataset $\{Y^{[1]}, \cdots, Y^{[N]}\}$ consisting of i.i.d. observations from $P_Y^\star$, and $\mathbf{Q}$ is assumed to be known. The goal is two-fold:

- **Unconditional generation**: to generate new samples from the ground-truth data distribution $P_X^\star$ approximately.

- **Posterior sampling**: to sample $X \sim P^\star(\cdot|Y)$ given an observation $Y$.

With this setup, the primary focus of recent work (Daras et al., 2023b;a; Rozet et al., 2024; Bai et al., 2024; Daras et al., 2024b) has been on **reconstruction** under a special class of likelihood functions. In such settings, the latent space is $\mathcal{X} = \mathbb{R}^{d_x}$ (consisting of "clean images"), and there is a known

distribution $P_A$ of corruption matrices $A \in \mathbb{R}^{d \times d_x}$. The observation is drawn as

$$Y = (AX + \epsilon, A), \quad \text{where } X \sim P_X^\star, A \sim P_A, \epsilon \sim \mathsf{N}\big(0, \sigma_Y^2 \mathbf{I}\big), \tag{2}$$

i.e., the observation $Y \in \mathbb{R}^d \times \mathbb{R}^{d \times d_x}$ is a (corrupted image, corruption matrix) pair, with the image corrupted by the matrix $A \sim P_A$ and the additive Gaussian noise $\epsilon$. By choosing different distributions $P_A$ for the corruption matrix, (2) can model problems including random masking (Daras et al., 2023b; Rozet et al., 2024; Bai et al., 2024) and blurring (Bai et al., 2024).

**Diffusion models.** Given samples from a data distribution $p_0$ over $\mathbb{R}^d$, diffusion models aim to learn how to generate new samples from $p_0$. Following Song et al. (2020), we consider the diffusion process $(X_t)_{t \in [0,1]}$ with $X_0 \sim p_0$, and $X_t | X_0 \sim \mathsf{N}\big(X_0, \sigma_t^2 \mathbf{I}\big)$. Formally, the diffusion process can be described by the following stochastic differential equation (SDE):

$$\mathrm{d}X_t = g(t)\mathrm{d}\mathbf{B}_t, \qquad X_0 \sim p_0, \tag{3}$$

where $g(t)^2 = \frac{d\sigma_t^2}{dt}$, and $(\mathbf{B}_t)_{t \in [0,1]}$ is the standard Brownian motion. Let $p_t(x)$ be the density function of $X_t \in \mathbb{R}^d$. It is well-known that the reverse of process (3) can be described by the following reverse-time diffusion process:

$$\mathrm{d}X_t = -g(t)^2 \nabla_x p_t(X_t)\mathrm{d}t + g(t)\mathrm{d}\mathbf{B}_t, \qquad X_1 \sim p_1. \tag{4}$$

With $\sigma_1$ being sufficiently large, we have $p_1 \approx \mathsf{N}\big(0, \sigma_1^2 \mathbf{I}\big)$. The score function $(x, t) \mapsto \nabla_x \log p_t(x)$ is typically parametrized by a neural network $\mathbf{s}_\theta(x, t)$. By Tweedie's formula, $\nabla_x \log p_t(x) = \frac{\mathbb{E}[X_0 | X_t = x] - x}{\sigma_t^2}$, where the expectation is taken with respect to the diffusion process (3). Hence, $\mathbf{s}_\theta(x, t)$ can be learned by optimizing the score-matching loss.

## 2 EXPECTATION-MAXIMIZATION APPROACH

When applied to our setup, the Expectation-Maximization (EM) method optimizes over a class of parameterized latent variable models $\{q_\theta(x, y)\}_\theta$ that aims to represent the joint ground-truth distribution of $(X, Y)$. Here, $q_\theta(x, y) : \mathcal{X} \times \mathcal{Y} \to \mathbb{R}_{\geq 0}$ is the probability density function associated with the model parametrized by parameter $\theta$, and we denote $q_\theta(y) : \mathcal{Y} \to \mathbb{R}_{\geq 0}$ to be the probability density function of the marginal distribution of the observable $Y$. EM seeks a parameter $\theta$ that maximizes the population log-likelihood of the observable variable:

$$\max_\theta \mathcal{L}(\theta) := \mathbb{E}_{Y \sim P_Y^\star} \log q_\theta(Y).$$

This optimization problem is equivalent to minimizing the KL divergence between $P_Y^\star$ and $q_\theta(y)$. However, direct optimization is computationally intractable for most problems. To overcome this computational challenge, each step of the EM method optimizes the following ELBO lower bound with a parameter $\widehat{\theta}$:

$$\mathcal{L}(\theta) \geq \mathbb{E}_{Y \sim P_Y^\star} \mathbb{E}_{X \sim q_{\widehat{\theta}}(X|Y)} \log \frac{q_\theta(X, Y)}{q_{\widehat{\theta}}(X|Y)}.$$

In particular, the EM algorithm can be succinctly written as: Starting from an initial point $\theta^{(0)}$, iterate

$$\theta^{(k+1)} = \arg\max_\theta \mathbb{E}_{Y \sim P_Y^\star} \mathbb{E}_{X \sim q_{\theta^{(k)}}(X|Y)} \log q_\theta(X, Y).$$

In our setting, since the likelihood $\mathbf{Q}$ is known and simple, the parametrized model should satisfy $q_\theta(x, y) = \mathbf{Q}(Y = y | X = x) q_\theta(x)$. In this case, the EM iterations reduce to

$$\theta^{(k+1)} = \arg\max_\theta \mathbb{E}_{Y \sim P_Y^\star} \mathbb{E}_{X \sim q_{\theta^{(k)}}(X|Y)} \log q_\theta(X). \tag{5}$$

This specialization of EM has been studied in (Aubin-Frankowski et al., 2022; Rozet et al., 2024; Bai et al., 2024), and it is also the basis of our framework. To simplify the notation, we consider the *mixture posterior distribution* $\pi^{(k)}$ with density $\pi^{(k)}(x) = \mathbb{E}_{Y \sim P_Y^\star}[q_{\theta^{(k)}}(x|Y)]$, which is a mixture with respect to the observation distribution $P_Y^\star$ of the posteriors $q_{\theta^{(k)}}(X|Y)$ (Rozet et al., 2024). Then, the EM update (5) can be rewritten as

$$\theta^{(k+1)} = \arg\min_\theta D_{\mathrm{KL}}\big(\pi^{(k)}(x) \,\|\, q_\theta(x)\big), \tag{6}$$

i.e., the model $q_{\theta^{(k+1)}}$ minimizes the distance to the mixture posterior distribution $\pi^{(k)}$. Crucially, to implement this update, we need to be able to sample from the conditional distribution $q_{\theta^{(k)}}(X|Y)$.

### 2.1 PRIOR APPROACH: EM WITH DIFFUSION PRIORS

In this section, we briefly review how prior work (Rozet et al., 2024; Bai et al., 2024) performs posterior sampling with diffusion models as priors. Their methods are restricted to the *linear corruption model* (2), where the observation is $Y = (AX + \epsilon, A)$, where $\epsilon \sim \mathsf{N}(0, \sigma_Y^2 \mathbf{I})$ is the noise and $A \sim P_A$ is a random corruption matrix. For simplicity, to describe these results, we focus on the case where $A$ is fixed, i.e. $\mathbf{Q}(\cdot|X) = \mathbf{Q}_A(\cdot|X) = \mathsf{N}(AX, \sigma_Y^2 \mathbf{I})$.

In the EM approach of Rozet et al. (2024); Bai et al. (2024), the latent variable models are described by diffusion models. More precisely, each $\theta$ parametrizes a score function $\mathbf{s}_\theta(x, t)$, and $q_\theta(x)$ corresponds to the distribution of $X_0$ obtained by running the backward diffusion process with the score function $\mathbf{s}_\theta$. However, to sample from $q_\theta(X|Y)$, one needs to approximate the conditional score function $\nabla_x \log q_\theta(X_t = x|Y = y)$. Following previous work on posterior sampling with diffusion priors (Chung et al., 2022, etc.), the conditional score is decomposed according to Bayes' rule:

$$\nabla_x \log q_\theta(X_t = x|Y) = \nabla_x \log q_\theta(Y|X_t = x) + \nabla_x \log q_\theta(X_t = x).$$

The second term is given by the score function $\mathbf{s}_\theta(x, t)$. To approximate the first term, Rozet et al. (2024) applies a Gaussian approximation $q_\theta(X = \cdot|X_t = x) \approx \mathsf{N}(\mathbb{E}_\theta[X|X_t = x], \mathbb{V}_\theta[X|X_t = x])$. Consequently, the conditional distribution of $Y$ is approximately

$$q_\theta(Y = \cdot|X_t = x) \approx \mathsf{N}(A\mathbb{E}_\theta[X|X_t = x], \sigma_Y^2 \mathbf{I} + A\mathbb{V}_\theta[X|X_t = x]A^\top).$$

Then, to calculate $\nabla_x \log q_\theta(Y|X_t = x)$, Rozet et al. (2024) introduces moment matching techniques to approximate the variance function $\mathbb{V}_\theta[X|X_t = x]$. Alternatively, Bai et al. (2024) applies a simpler approximation $q_\theta(Y = \cdot|X_t = x) \approx \mathsf{N}(A\mathbb{E}_\theta[X|X_t = x], \sigma_Y^2 \mathbf{I})$.

However, these approximations all rely on the assumption that $q_\theta(X_0 = \cdot|X_t = x)$ is close to a Gaussian distribution. This assumption may not hold for general diffusion priors, which are highly multi-modal. Therefore, errors in these approximation schemes can be difficult to control. Furthermore, even when the learned diffusion prior $q_\theta$ is close to the ground truth, the posterior distribution of $X|Y$ (obtained by approximating the score $\nabla_x \log q_\theta(X_t = x|Y)$) might not accurately represent the true conditional distribution $q_\theta(X|Y)$ under the diffusion prior $q_\theta(X)$.

Additionally, the moment matching techniques of Rozet et al. (2024) are rather sophisticated and specialized to (2). For general likelihood with non-linear transformations, calculating the score $\nabla_x \log q_\theta(Y|X_t = x)$ can be challenging even under the Gaussian approximation assumption.

### 2.2 OUR APPROACH: EM WITH CONDITIONAL DIFFUSION MODEL

Instead of parametrizing the data distribution $q_\theta(x)$ using a diffusion model, we directly model the posterior distribution $q_\theta(x|y)$ through a conditional score function network $\mathbf{s}_\theta(x, t|y)$. Below, we describe the corresponding conditional diffusion process for generating posterior samples.

**Conditional diffusion process.** Given a latent variable model $q$, we consider the diffusion process

$$(X_0, Y) \sim q, \qquad \mathrm{d}X_t = g(t)\mathrm{d}\mathbf{B}_t. \tag{7}$$

Let $p$ be the joint distribution of $(\{X_t\}_{t \in [0,1]}, Y)$. To sample from $q(X_0|Y)$, we consider the following reverse-time process:

$$\mathrm{d}X_t = -g(t)^2 \mathbf{s}_\theta(X_t, t|Y)\mathrm{d}t + g(t)\mathrm{d}\mathbf{B}_t, \qquad X_1 \sim q_1(\cdot|Y), \tag{8}$$

where the network $\mathbf{s}_\theta$ directly approximates the true conditional score function

$$\mathbf{s}_\theta(x, t|y) \approx \nabla_x \log p(X_t = x|y) = \frac{\mathbb{E}[X_0|X_t = x, Y = y] - x}{\sigma_t^2}, \tag{9}$$

and the expectation is taken over the process (7) (see e.g. (Daras et al., 2024a)). For a given parameter $\theta$ that parametrizes the conditional denoiser network $\mathbf{s}_\theta$, we let $q_\theta(X = \cdot|Y)$ be the distribution of $X_0$ generated by (8). In particular, when $\mathbf{s}_\theta(x, t|y) = \nabla_x \log p(X_t = x|y)$, the reverse process (8) indeed generates $X_0 \sim q(\cdot|Y)$, i.e., $q_\theta(\cdot|Y) = q(\cdot|Y)$.

---

**Algorithm 1** DiffEM: Expectation-Maximization with a conditional diffusion model

---

**Input:** Dataset of corrupted observations $\mathcal{D}_Y = \left\{ Y^{[1]}, \cdots, Y^{[N]} \right\}$, likelihood $\mathbf{Q}(\cdot|X)$, and a initialization for the conditional model $\theta^{(0)}$.

    **for** $k = 0, 1, \cdots, K-1$ **do**

        **// E-step:**

        **for** $i \in [N]$ **do**

            Generate the reconstruction $X^{[i]} \sim q_{\theta^{(k)}}(\cdot|Y^{[i]})$ using the current conditional model $\theta^{(k)}$.

        **// M-step:**

        Train a new conditional diffusion model using the dataset $\mathcal{D}_X^{(k)} = \{X^{[1]}, \cdots, X^{[N]}\}$ by minimizing the objective provided in (10):

$$\theta^{(k+1)} = \arg\min_\theta L_{\text{SM},k}(\theta).$$

**Output:** (1) The conditional diffusion model $\theta^{(K)}$, and

**Output:** (2) An *unconditional* diffusion model $\widehat{\theta}$ trained on the dataset $\mathcal{D}_X^{(K-1)}$.

---

**EM with conditional diffusion models.** Based on the conditional diffusion process, we propose the EM procedure Algorithm 1, using a conditional diffusion model to learn the *posterior* directly. In the E-step, the algorithm generates the dataset $\mathcal{D}_X^{(k)} = \{X^{[1]}, \cdots, X^{[N]}\}$ consisting of the reconstruction $X^{[i]} \sim q_{\theta^{(k)}}(\cdot|Y^{[i]})$. Then, in the M-step, the algorithm uses the dataset $\mathcal{D}_X^{(k)}$ to train the conditional diffusion model $\theta^{(k+1)}$, so that it learns to sample from $\widehat{P}^{(k)}(X|Y)$, the posterior of $\widehat{P}^{(k)}(X,Y)$ which samples $X \sim \mathcal{D}_X^{(k)}$ and then samples $Y \sim \mathbf{Q}(\cdot|X)$. To train this model, we consider the following conditional score matching loss:

$$L_{\text{SM},k}(\theta) = \int_0^1 \lambda_t \mathbb{E}_{X \sim \mathcal{D}_X^{(k)}, Y \sim \mathbf{Q}(\cdot|X)} \mathbb{E}_{X_t = X + \sigma_t Z} \left\| \mathbf{s}_\theta(X_t, t|Y) + Z \right\|^2 \mathrm{d}t, \qquad (10)$$

where $Z \sim \mathsf{N}(0, \mathbf{I})$ is the unit noise, and $\lambda_t \geq 0$ is a weight sequence. It is straightforward to verify that, assuming the network $\mathbf{s}_\theta$ is expressive enough, the minimizer $\theta^\star$ of $L_{\text{SM},k}$ satisfies $\mathbf{s}_{\theta^\star}(x, t|y) = \frac{\mathbb{E}[X_0|X_t = x, Y = y] - x}{\sigma_t^2}$, where the conditional expectation is taken with respect to the distribution sampling variables as $X_0 \sim \mathcal{D}_X^{(k)}$, $Y \sim \mathbf{Q}(\cdot|X_0)$, $X_t \sim \mathsf{N}(X_0, \sigma_t^2 \mathbf{I})$. Therefore, as long as the M-step is done successfully, we expect to have $q_{\theta^{(k+1)}}(X|Y) \approx \widehat{P}^{(k)}(X|Y)$ (cf. Section 3).

**The advantage of conditional diffusion model.** Unlike approaches that rely on ad hoc approximation schemes for the posterior score function using unconditional diffusion models (Rozet et al., 2024; Bai et al., 2024), our framework directly employs a conditional diffusion model. Both the *data distribution* and the *likelihood function* are implicitly encoded in this model through the minimization of the conditional score matching loss (10). In experiments (Section 4), we observe that DiffEM consistently outperforms EM methods with diffusion priors. As predicted by our theoretical analysis (Section 3), this improvement is largely due to the fact that conditional models avoid the approximation bottleneck inherent in heuristic posterior sampling schemes.

**Output: Posterior sampler $\theta^{(K)}$ and diffusion prior $\widehat{\theta}$.** Our framework is designed to address two complementary goals: (1) posterior sampling and (2) unconditional generation (cf. Section 1.1). The conditional diffusion model trained by DiffEM naturally serves as a posterior sampler. For unconditional generation, we leverage the reconstructed dataset $\mathcal{D}_X^{(K-1)}$ generated during the final EM iteration, and train an unconditional diffusion prior on this dataset. In particular, when the target application requires only a diffusion prior (Daras et al., 2023b; Rozet et al., 2024; Bai et al., 2024), we may directly use $\widehat{\theta}$. In such cases, the conditional model adopted by our approach primarily serves as a means to accelerate EM convergence.

**Computational efficiency of DiffEM.** The computational cost of DiffEM can be decomposed as

$$\text{Total Time} = T_{\text{init}} + K \cdot T_{\text{ft}} + T_{\text{u}}, \qquad (11)$$

where $K$ is the number of EM iterations, $T_{\text{init}}$ is the time of training a standard conditional diffusion model from scratch, $T_{\text{ft}} \leq T_{\text{init}}$ is the average time of *fine-tuning* the conditional diffusion model for each M-step, and $T_{\text{u}}$ is the cost of training an unconditional model to output. The cost $T_{\text{init}} \geq T_{\text{ft}}$

of training diffusion model is intrinsic to diffusion-based learning methods. Thus, DiffEM can be interpreted as increasing the training cost by a multiplicative factor of $K$ (the number of EM iterations), which we view as the *unavoidable* cost of working with only corrupted data.

In general, the computational cost of EM-based methods (Rozet et al., 2024; Bai et al., 2024) can always be decomposed as (11). In our experiments, we compare the computation time $K$, $T_{\mathsf{init}}$, and $T_{\mathsf{ft}}$ of DiffEM and EM-MMPS in CIFAR-10 experiments (Table 3).

## 3 MONOTONIC IMPROVEMENT PROPERTY AND CONVERGENCE

In this section, we analyze the convergence properties of the EM iteration. As observed by Aubin-Frankowski et al. (2022), when the iteration (6) is *exact*, i.e., when the sample size is infinite and the conditional model $q_{\theta^{(k+1)}}$ learns the mixture posterior exactly in each M-step, the EM iteration is equivalent to *mirror descent* in the space of measures. Therefore, the convergence of the *exact* EM iteration follows immediately from the guarantees of mirror descent.

We study the DiffEM iteration, taking the *score-matching error* introduced by the M-step into account. For simplicity, we analyze the EM iteration with *fresh* corrupted samples. Specifically, we consider the variant of Algorithm 1 where, at each iteration $k = 0, 1, \cdots, K-1$, a new dataset of corrupted observations $\mathcal{D}_Y^{(k)} = \{Y^{[1]}, \cdots, Y^{[N]}\} \sim P_Y^\star$ is drawn in the E-step. We continue to refer to this procedure as DiffEM throughout this section.

Under this variant, for each $k$, the reconstructed dataset $\mathcal{D}_X^{(k)} = \{X^{[1]}, \cdots, X^{[N]}\}$ consists of i.i.d samples from the posterior mixture distribution $\pi^{(k)} = \mathbb{E}_{Y \sim P_Y^\star}[q_{\theta^{(k)}}(\cdot|Y)]$. We let $P^{(k)}$ be the joint probability distribution of $(X, Y)$ under $X \sim \pi^{(k)}, Y \sim \mathbf{Q}(\cdot|X)$, and write $P_Y^{(k)}$ for the marginal of $Y$. The convergence is measured in terms of $D_{\mathrm{KL}}(P_Y^\star \parallel P_Y^{(k)})$, the Kullback-Leibler (KL) divergence between the true observation distribution $P_Y^\star$ and the distribution $P_Y^{(k)}$. Intuitively, this measures how plausible the prior $\pi^{(k)}$ is by comparing the induced observation distribution $P_Y^{(k)}$ to $P_Y^\star$. [1]

**Score-matching error.** We define the *score-matching error* of the $k$th M-step as

$$\varepsilon_{\mathsf{SM}}^{(k)} := \mathbb{E}_{Y \sim P_Y^\star} D_{\mathrm{KL}}(q_{\theta^{(k+1)}}(\cdot|Y) \parallel P^{(k)}(\cdot|Y)),$$

which measures the KL divergence between the conditional diffusion model $q_{\theta^{(k+1)}}$ learned in the $k$th M-step and the true posterior $P^{(k)}(\cdot|Y)$. This error can be decomposed into two components: (1) the error of the learned score function, which is the statistical error of score matching (10) with a finite sample size, and (2) the sampling error, which comes from the discretized backward diffusion process (8) starting from a noisy Gaussian. When the denoiser network is sufficiently expressive, the score matching error can be upper bounded through statistical learning theory (Dou et al., 2024; Zhang et al., 2024; Wibisono et al., 2024; Chen et al., 2024; Gatmiry et al., 2024, etc.). The sampling error is addressed by existing work on backward diffusion sampling (see e.g., Chen et al., 2022; Conforti et al., 2023; 2025)). Therefore, under appropriate conditions, it can be shown that the score-matching error $\varepsilon_{\mathsf{SM}}^{(k)} \to 0$ as the sample size $N$ increases.

**Monotonicity of EM.** Our first result (shown in Appendix B.1) is the following approximate *monotonicity* property of the EM iteration in terms of the statistical error $\varepsilon_{\mathsf{SM}}^{(k)}$.

**Lemma 1** (Monotonic improvement). *For any $k \geq 0$, it holds that*

$$\underbrace{D_{\mathrm{KL}}(P_Y^\star \parallel P_Y^{(k+1)})}_{\text{error of prior } \pi^{(k+1)}} \leq \underbrace{D_{\mathrm{KL}}(P_Y^\star \parallel P_Y^{(k)})}_{\text{error of prior } \pi^{(k)}} - \underbrace{D_{\mathrm{KL}}(\pi^{(k+1)} \parallel \pi^{(k)})}_{\text{difference between priors}} + \underbrace{\varepsilon_{\mathsf{SM}}^{(k)}}_{\text{score-matching error of } q_{\theta^{(k+1)}}}.$$

Therefore, when the statistical error $\varepsilon_{\mathsf{SM}}^{(k)} \to 0$, the divergence $D_{\mathrm{KL}}(P_Y^\star \parallel P_Y^{(k)})$ is monotonically decreasing. In other words, in the EM iteration, the observation distribution induced by prior $\pi^{(k+1)}$ is always closer to $P_Y^\star$ compared to the observation distribution induced $\pi^{(k)}$, modulo the score-matching error $\varepsilon_{\mathsf{SM}}^{(k)}$. In Section 4.1.1, we corroborate this property in experiments, showing that DiffEM can improve upon the learned prior produced by EM-MMPS (Rozet et al., 2024).

---

[1]Here, the convergence is not measured as the divergence between the data distribution $P_X^\star$ and $\pi^{(k)}$ because in general, the problem (1) might not be *identifiable*, i.e., there can exist a distribution $P_X' \neq P_X^\star$ that induces the same observation distribution $\mathbf{Q}_\# P_X' = P_Y^\star$. Therefore, convergence of the data distribution can only be obtained under the additional assumption of *identifiability* (cf. Assumption 1).

**Convergence rate.** Beyond monotonicity, we show that the EM iteration enjoys a convergence rate guarantee. However, this guarantee requires that the conditional model achieves small approximation error measured in the latent space. Specifically, for each $k \geq 0$, we define the error

$$\widetilde{\varepsilon}_{\mathsf{SM}}^{(k)} = \mathbb{E}_{(X,Y) \sim P^\star} \left[ \log \frac{P^{(k)}(X|Y)}{q_{\theta^{(k+1)}}(X|Y)} \right],$$

which measures the closeness of the posterior likelihoods computed under $P^{(k)}$ and $q_{\theta^{(k+1)}}$ with respect to samples $(X, Y) \sim P^\star$. The error $\widetilde{\varepsilon}_{\mathsf{SM}}^{(k)}$ can be larger than the $\varepsilon_{\mathsf{SM}}^{(k)}$ since it is measured under the unknown prior distribution $P_X^\star$. Nevertheless, we show that $\widetilde{\varepsilon}_{\mathsf{SM}}^{(k)}$ can be related to $\varepsilon_{\mathsf{SM}}^{(k)}$ under appropriate assumptions (detailed in Appendix B.4). Below, we state the convergence guarantee of the EM iteration. The proof is in Appendix B.2.

**Proposition 2** (Convergence of EM iteration). *For each $K \geq 0$, we have*

$$\min_{k \leq K} D_{\mathrm{KL}}(P_Y^\star \,\|\, P_Y^{(k)}) \leq \frac{1}{K+1} \sum_{i=0}^{K} D_{\mathrm{KL}}(P_Y^\star \,\|\, P_Y^{(k)}) \leq \frac{D_{\mathrm{KL}}(P_X^\star \,\|\, \pi^{(0)})}{K+1} + \max_{k \leq K} \widetilde{\varepsilon}_{\mathsf{SM}}^{(k)}.$$

Therefore, as the number of EM iterations increases, $P_Y^{(k)}$ converges to $P_Y^\star$ at the rate of $\frac{1}{k}$, up to the statistical error $\widetilde{\varepsilon}_{\mathsf{SM}}^{(k)}$. Furthermore, we can also derive the following last-iterate convergence by invoking Lemma 1:

$$D_{\mathrm{KL}}(P_Y^\star \,\|\, P_Y^{(K)}) \leq \frac{D_{\mathrm{KL}}(P_X^\star \,\|\, \pi^{(0)})}{K+1} + \max_{k \leq K} \widetilde{\varepsilon}_{\mathsf{SM}}^{(k)} + \sum_{k=0}^{K} \varepsilon_{\mathsf{SM}}^{(k)}, \qquad \forall K \geq 0.$$

Given that each EM update is computationally expensive, the above convergence rate is most relevant in the regime where $D_{\mathrm{KL}}(P_X^\star \,\|\, \pi^{(0)}) \lesssim 1$, i.e., where the initial diffusion model provides a prior that is not too far from the ground-truth $P_X^\star$. Such a *warm start* model can be trained using existing methods (Daras et al., 2023b) that are computationally cheaper.

**Stronger convergence under identifiability.** Under the assumption that the latent variable problem (1) is *identifiable*, we show that EM achieves *linear* convergence in terms of $D_{\mathrm{KL}}(P_X^\star \,\|\, \pi^{(k)})$.

**Assumption 1** (Identifiability). *There exists parameter $\kappa \geq 1, R \geq 0$ such that for any distribution $P(x)$ with $D_{\mathrm{KL}}(P_X^\star \,\|\, P) \leq R$, it holds that*

$$D_{\mathrm{KL}}(P_X^\star \,\|\, P) \leq \kappa \cdot D_{\mathrm{KL}}(P_Y^\star \,\|\, \mathbf{Q}_\# P),$$

*where $\mathbf{Q}_\# P$ is the distribution of $Y$ under $X \sim P, Y \sim \mathbf{Q}(\cdot|X)$.*

In other words, Assumption 1 requires that for any prior $P$ whose induced observation distribution $\mathbf{Q}_\# P$ is close to $P_Y^\star$, $P$ itself must be close to the true data distribution $P_X^\star$. Intuitively, Assumption 1 quantifies the *identifiability* of the latent variable problem (1). We show the following in Appendix B.3.

**Proposition 3** (Linear convergence of EM). *Suppose that Assumption 1 holds, $D_{\mathrm{KL}}(P_X^\star \,\|\, \pi^{(0)}) \leq R$, and $\widetilde{\varepsilon}_{\mathsf{SM}}^{(k)} \leq \frac{R}{\kappa}$ for each $k \geq 0$. Then it holds that*

$$D_{\mathrm{KL}}(P_X^\star \,\|\, \pi^{(K)}) \leq \exp\left(-\frac{K}{\kappa+1}\right) \cdot D_{\mathrm{KL}}(P_X^\star \,\|\, \pi^{(0)}) + (\kappa + 1) \max_k \widetilde{\varepsilon}_{\mathsf{SM}}^{(k)}.$$

## 4 EXPERIMENTS

We evaluate the proposed method, DiffEM, through a series of experiments. We begin with a synthetic manifold learning task (Appendix C.1), where we show that the conditional diffusion model yields more accurate posterior samples than existing approximate posterior sampling schemes (Rozet et al., 2024). We then conduct distributional learning and image reconstruction experiments on CIFAR-10 (Section 4.1) and CelebA (Section 4.2), demonstrating that DiffEM outperforms prior approaches for learning diffusion models from corrupted data.

### 4.1 CORRUPTED CIFAR-10

We next evaluate our method on the CIFAR-10 dataset (Krizhevsky, 2009), treating the 50000 training images as samples from the latent distribution $P_X^\star$.

| Task | Method | IS ↑ | FID ↓ | FD$_{\text{DINOv2}}$ ↓ | FD$_\infty$ ↓ |
|---|---|---|---|---|---|
| Posterior Sampling | Ambient-Diffusion | 7.70 | 30.76 | 260.23 | 256.11 |
| | EM-MMPS | 9.77 | 6.49 | 237.02 | 231.80 |
| | DiffEM (Ours) | **9.81** | **4.68** | **220.97** | **216.53** |
| | DiffEM (Warm-started) | 9.66 | **4.66** | **186.90** | **180.70** |
| Unconditional Generation | Ambient-Diffusion | 6.88 | 28.88 | 1068.00 | 1062.98 |
| | EM-MMPS | 8.14 | 13.18 | 643.59 | 640.14 |
| | DiffEM (Ours) | **8.57** | **10.24** | **598.18** | **594.75** |
| | DiffEM (Warm-started) | 8.49 | 10.33 | **546.07** | **541.53** |

Table 1: Posterior sampling and unconditional generation performance on CIFAR-10 with random masking with corruption rate of $\rho = 0.75$ compared to Ambient-Diffusion (Daras et al., 2023b) and EM-MMPS (Rozet et al., 2024). The details of DiffEM with warm-start are described in Section 4.1.1.

| Task | Method | density | recall | precision | coverage |
|---|---|---|---|---|---|
| Posterior Sampling | Ambient-Diffusion | 0.87616 | 0.75420 | **0.79210** | 0.67930 |
| | EM-MMPS | 0.68918 | 0.83780 | 0.72770 | 0.67160 |
| | DiffEM (Ours) | 0.58080 | **0.87110** | 0.70150 | 0.64080 |
| | DiffEM (Warm-started) | 0.72216 | 0.86300 | 0.76320 | **0.72490** |
| Unconditional Generation | Ambient-Diffusion | 1.40812 | 0.0825 | **0.79370** | 0.08170 |
| | EM-MMPS | 0.80986 | 0.4895 | 0.64740 | 0.24380 |
| | DiffEM (Ours) | 0.81284 | 0.50490 | 0.64900 | 0.25640 |
| | DiffEM (Warm-started) | 0.75816 | **0.52560** | 0.65980 | **0.29370** |

Table 2: Additional metrics (density, recall, precision and coverage) following the work (Stein et al., 2023) for posterior sampling and unconditional generation on CIFAR-10 with random masking at corruption rate $\rho = 0.75$, complementing the main quantitative evaluations comparing DiffEM, EM-MMPS (Rozet et al., 2024), and Ambient-Diffusion (Daras et al., 2023b).

**Masked corruption.** Following (Daras et al., 2023b; Rozet et al., 2024), we consider randomly masking each pixel with probability $\rho$, i.e., the matrix $A \sim P_A$ in (2) is diagonal with entries independently drawn from Bernoulli$(1 - \rho)$. In this setting, the observation is generated as $Y = (AX + \epsilon, A)$, with $A \sim P_A$, $X \sim P_X^\star$, $\epsilon \sim \mathsf{N}(0, \sigma_Y^2 \mathbf{I})$. In other words, each image is corrupted by (1) first randomly deleting every pixel independently with probability $\rho$, and then (2) adding isotropic Gaussian noise with variance $\sigma_Y^2$.

In our experiments, we set $\rho = 0.75$, $\sigma_Y^2 = 10^{-6}$, i.e., each image has 75% of the pixels deleted and is corrupted by negligible Gaussian noise. We also perform experiments with corruption level $\rho = 0.9$ and report the results in Table 7.

**Experiment setup.** Our conditional diffusion model $q_\theta(x|y)$ is parametrized by a denoiser network $d_\theta(x_t, t, y)$ with U-net architecture. We train the model for 21 DiffEM iterations, initializing with a Gaussian prior (detailed in Appendix C). For each iteration, we train the denoiser network with conditional score matching (10) to learn the conditional mean $\mathbb{E}[X_0|X_t, Y]$. We then compare DiffEM to prior methods (Daras et al., 2023b; Rozet et al., 2024) under the following evaluation metrics, which correspond to the *posterior sampling* task and *unconditional generation* task (cf. Section 1.1).

**Eval 1: Posterior sampling performance.** The final model returned by DiffEM is a *conditional diffusion model*, i.e., given any corrupted observation $Y$, the model samples a reconstructed image $X \sim q_\theta(\cdot|Y)$. Therefore, to evaluate the performance of posterior sampling, for each observation $Y^{[i]}$ in our dataset, we use the trained model to generate a reconstructed image $X^{[i]} \sim q_\theta(\cdot|Y^{[i]})$ and obtain the reconstructed dataset $\mathcal{D}_{\text{recon}} = \{X^{[1]}, \cdots, X^{[50000]}\}$ (similar to the E-step of Algorithm 1). We then evaluate the quality of $\mathcal{D}_{\text{recon}}$ by computing the Inception Score (IS) (Salimans et al., 2016) and

| Method | $K$ | $T_{\mathsf{init}}$ | $T_{\mathsf{ft}}$ | $T_{\mathsf{u}}$ |
|--------|-----|------|------|-----|
| EM-MMPS | 32 | $43.0 \pm 0.8$ | $86.3 \pm 0.7$ | N/A |
| DiffEM | 21 | $63.5 \pm 0.4$ | $70.3 \pm 0.2$ | $74.54 \pm 0.09$ |

Table 3: Comparison of computation time (cf. Section 2.2), with $T_{\mathsf{init}}, T_{\mathsf{ft}}, T_{\mathsf{u}}$ measured in minutes using $4\times$ H200. The cost of EM-MMPS (Rozet et al., 2024) can similarly be decomposed as $T_{\mathsf{init}} + K \cdot T_{\mathsf{ft}}$ (it does not incur the cost $T_{\mathsf{u}}$). As shown, DiffEM is more computationally efficient.

the Fréchet distance to the uncorrupted dataset in various representation spaces[2] to obtain the metrics FID (Heusel et al., 2017), $FD_{DINOv2}$ (Oquab et al., 2023; Stein et al., 2023), and $FD_{\infty}$ (Chong and Forsyth, 2020). The results are reported in Table 1. Furthermore, we evaluate Precision, Coverage, Recall, and Density following (Stein et al. (2023)). The results are provided in Table 2.

**Eval 2: Unconditional generation performance.** We also note that the models trained by existing works (Daras et al., 2023b; Rozet et al., 2024; Bai et al., 2024) are *unconditional* diffusion models, which can be regarded as the reconstruction of the ground-truth data distribution $P_X$. In DiffEM, the reconstructed data distribution is implicitly described by the conditional diffusion model $q_\theta$. Therefore, to evaluate the data distribution recovered by DiffEM, we use the reconstructed dataset $\mathcal{D}_{\mathrm{recon}}$ to train a new (unconditional) diffusion model $p_{\theta_{\mathrm{uncond}}}$, which learns to sample from the data distribution induced by $q_\theta$. We then evaluate the metrics (IS, FID, $FD_{\infty}$, $FD_{DINOv2}$, Precision, Recall, Density, Coverage) of the model $p_{\theta_{\mathrm{uncond}}}$ as our performance on the *unconditional generation* task. We report the metrics in Table 1 and Table 2.

**Discussion and comparison.** We compare DiffEM to Ambient-Diffusion (Daras et al., 2023b)[3] and EM-MMPS (Rozet et al., 2024) under the above metrics in Table 1 (higher IS and lower FID/FD scores indicate better performance) and Table 2 (higher recall, precision and coverage indicates better performance). To evaluate the diffusion prior trained by these baselines, we apply their approximate posterior sampling scheme and report the metrics evaluated on the reconstructed dataset. Under all four metrics, the diffusion models trained by DiffEM outperform both Ambient-Diffusion and EM-MMPS, demonstrating the power of our pipeline.[4] Figure 6 shows qualitative results comparing the corrupted observations and reconstructions from our model.

We also compare the computational cost of DiffEM and EM-MMPS in Table 3 and in the Figure 2 following our discussion in Section 2.2.

### 4.1.1 DiffEM with warm-start

Additionally, we perform experiments on the masked CIFAR-10 dataset with *warm-started* DiffEM. Specifically, we take the diffusion prior trained by 32 iterations of EM-MMPS (Rozet et al., 2024), and perform 10 DiffEM iterations starting from this prior. We evaluate the final posterior sampling performance and unconditional generation quality (reported in Table 1 and Table 2).

The results show that using a high-quality initial prior accelerates the convergence of DiffEM: only 10 DiffEM iterations are needed. This observation is consistent with our theoretical results (Section 3). Furthermore, warm-started DiffEM outperforms DiffEM with an initial Gaussian prior in terms of the scores $FD_{DINOv2}$ and $FD_{\infty}$, indicating that DiffEM can converge to a better distribution when starting from an informed prior.[5] We also plot the evolution of the IS, FID, DINO, and $FD_{\infty}$ scores in Figure 9, which corroborates the monotonic improvement property of DiffEM (Lemma 1).

---

[2]The Fréchet distance measures discrepancies at the *distributional* level. Under severe corruption (75% of pixels deleted), the posterior distribution $P_{X|Y}^\star$ may not concentrate around a single ground-truth. As a result, classical reconstruction metrics such as PSNR and LPIPS are less appropriate in this setting (Rozet et al., 2024).

[3]We note that the Ambient-Diffusion model was trained on a dataset with corruption level $\rho' = 0.6$, an easier setting than ours ($\rho = 0.75$).

[4]We note that Bai et al. (2024) proposed EM-Diffusion and reported FID score 21.08 (corruption level $\rho' = 0.6$ and initialized with a diffusion prior trained on 50 clean images). However, we cannot reproduce their experiments to evaluate other metrics. Given that EM-MMPS (Rozet et al., 2024) achieves a much better FID score than EM-Diffusion (Bai et al., 2024), we believe it is sufficient to compare DiffEM to EM-MMPS.

[5]However, it is worth noting that warm-started DiffEM is computationally more expensive, as the warm-start prior requires training with 32 iterations of EM-MMPS.

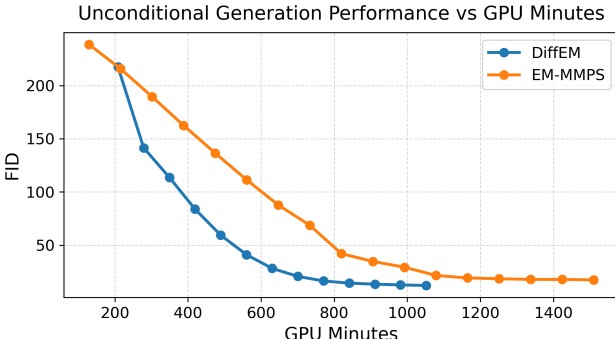

Figure 2: Evolution of the performance of EM-MMPS and DiffEM as a function of GPU hours. Both methods are trained on 4×H200 GPUs. DiffEM converges substantially faster and achieves superior performance for any fixed compute budget.

| Task | $\rho$ | Method | IS $\uparrow$ | FID $\downarrow$ | FD$_{\mathrm{DINOv2}}$ $\downarrow$ | FD$_{\infty}$ $\downarrow$ |
|---|---|---|---|---|---|---|
| Posterior sampling | 0.5 | EM-MMPS | 3.237 | 0.61 | 9.36 | 6.07 |
| | | DiffEM | **3.239** | **0.33** | **5.07** | **2.07** |
| | 0.75 | EM-MMPS | 2.96 | 31.22 | 113.09 | 109.41 |
| | | DiffEM | **3.16** | **1.43** | **39.34** | **36.26** |
| Unconditional generation | 0.5 | EM-MMPS | 2.50 | 11.44 | **186.16** | **182.90** |
| | | DiffEM | **2.52** | **10.11** | 344.60 | 340.97 |
| | 0.75 | EM-MMPS | 2.35 | 61.40 | **321.90** | **319.58** |
| | | DiffEM | **2.50** | **10.75** | 423.95 | 420.76 |

Table 4: Performance of DiffEM and EM-MMPS (Rozet et al., 2024) on masked CelebA with masking probability $\rho \in \{0.5, 0.75\}$.

### 4.1.2 ADDITIONAL EXPERIMENT: CIFAR-10 UNDER GAUSSIAN BLUR

In addition to the masked corruption experiment, we perform experiments on the *blurred CIFAR-10* dataset. In the Gaussian blur model, each observation $Y \sim \mathsf{N}(AX, \sigma_Y^2)$ is generated by applying a Gaussian blur kernel on $X$ with standard deviation $\sigma_{\mathrm{kernel}}$ (represented by the matrix $A$), and then adding isotropic Gaussian noise $\epsilon \sim \mathsf{N}(0, \sigma_Y^2 \mathbf{I})$. In the experiment, we set $\sigma_{\mathrm{kernel}} = 2$ and $\sigma_Y^2 = 10^{-6}$ and follow the same training procedure as in the masked CIFAR-10 experiment (details in Appendix C.3).

### 4.2 CORRUPTED CELEBA

We perform experiments on the CelebA dataset (Liu et al., 2018), with images cropped to $64 \times 64$ pixels following (Wang et al., 2023; Daras et al., 2023b). We consider the setting in Section 4.1 with masking probability $\rho \in \{0.5, 0.75\}$ and noise level $\sigma_Y^2 = 0$, i.e., the corruption level is moderate. We initialize the first iteration for DiffEM with the Gaussian prior (cf. Appendix C). We evaluate the diffusion models trained by DiffEM following the protocol of Section 4.1 (Table 4). As shown in Table 4, DiffEM significantly outperforms EM-MMPS. We also present sample reconstructed images in Appendix C.11 and an illustration of the pipeline in Figure 1.

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

## A  RELATED WORK

**Learning diffusion models with corrupted datasets.**  Recent advances in diffusion models (Ho et al., 2020; Song et al., 2020) have demonstrated remarkable success in learning high-dimensional distributions. However, training diffusion models with corrupted data presents significant challenges, as most existing techniques are designed for clean datasets, and learning latent variable models is known to be theoretically and practically difficult. Several approaches have been proposed to address this challenge using diffusion models. For linear corruption $Y \sim \mathsf{N}\left(AX, \sigma_Y^2 \mathbf{I}\right)$ (cf. Eq. (2)), methods such as SURE-score (Aali et al., 2023), GSURE (Kawar et al., 2023), and Ambient-Diffusion (Daras et al., 2023b; Aali et al., 2025; Daras et al., 2025a) train the denoiser network using a surrogate loss function. Specializing to Gaussian corruption $Y \sim \mathsf{N}\left(X, \sigma_Y^2 \mathbf{I}\right)$, Daras et al. (2023a; 2024b) propose enforcing *consistency* of the diffusion model to enable generalization to unseen noise levels, while Lu et al. (2025) develop an iterative scheme to refine the diffusion prior. Recent work (Rozet et al., 2024; Bai et al., 2024) identifies the Expectation-Maximization (EM) method as a promising framework for training diffusion priors with linearly corrupted observations. However, as these EM approaches employ diffusion models as *priors*, they rely heavily on approximation schemes for posterior sampling (detailed discussion in Section 2.1).

**Solving inverse problems with diffusion models.**  Diffusion models have also been shown as powerful priors for a wide range of inverse problems in computer vision and medical imaging. A line of work—including SNIPS (Kawar et al., 2021), ILVR (Choi et al., 2021), DDRM (Kawar et al., 2022), Palette (Saharia et al., 2022), and DPS (Chung et al., 2022), among others—has demonstrated the effectiveness of both unconditional and conditional diffusion models in addressing various tasks, such as super-resolution, inpainting, deblurring, and compressed sensing. As surveyed by Daras et al. (2024a), many of these approaches leverage learned diffusion priors and perform posterior sampling through approximations of the posterior score function, and the previous work on EM (Rozet et al., 2024; Bai et al., 2024) also follows this approach.

# B  PROOFS FROM SECTION 3

## B.1  PROOF OF LEMMA 1

Note that

$$D_{\mathrm{KL}}(P_Y^\star \parallel P_Y^{(k)}) - D_{\mathrm{KL}}(P_Y^\star \parallel P_Y^{(k+1)}) = \mathbb{E}_{Y \sim P_Y^\star} \log \frac{P_Y^{(k+1)}(Y)}{P_Y^{(k)}(Y)}.$$

By definition and Bayes' rule,

$$\begin{aligned}
P_Y^{(k+1)}(y) &= \int \mathbf{Q}(y|x)\pi^{(k+1)}(x)\mathrm{d}x = \int \mathbf{Q}(y|x)\pi^{(k)}(x) \cdot \frac{\pi^{(k+1)}(x)}{\pi^{(k)}(x)}\mathrm{d}x \\
&= \int P^{(k)}(x,y) \cdot \frac{\pi^{(k+1)}(x)}{\pi^{(k)}(x)}\mathrm{d}x \\
&= \int P_Y^{(k)}(y) \cdot P^{(k)}(x|y) \cdot \frac{\pi^{(k+1)}(x)}{\pi^{(k)}(x)}\mathrm{d}x \\
&= P_Y^{(k)}(y) \cdot \mathbb{E}_{X \sim q_{\theta^{(k+1)}}(\cdot|y)}\left[ \frac{P^{(k)}(X|y)}{q_{\theta^{(k+1)}}(X|y)} \cdot \frac{\pi^{(k+1)}(X)}{\pi^{(k)}(X)} \right].
\end{aligned}$$

Therefore, by Jensen's inequality, we have

$$\begin{aligned}
&D_{\mathrm{KL}}(P_Y^\star \parallel P_Y^{(k)}) - D_{\mathrm{KL}}(P_Y^\star \parallel P_Y^{(k+1)}) \\
&= \mathbb{E}_{Y \sim P_Y^\star} \log \frac{P_Y^{(k+1)}(Y)}{P_Y^{(k)}(Y)} \\
&= \mathbb{E}_{Y \sim P_Y^\star} \log \mathbb{E}_{X \sim q_{\theta^{(k+1)}}(\cdot|Y)}\left[ \frac{\pi^{(k)}(X|Y)}{q_{\theta^{(k+1)}}(X|Y)} \cdot \frac{\pi^{(k+1)}(X)}{\pi^{(k)}(X)} \right] \\
&\geq \mathbb{E}_{Y \sim P_Y^\star}\mathbb{E}_{X \sim q_{\theta^{(k+1)}}(\cdot|Y)}\left[ \log\left( \frac{P^{(k)}(X|Y)}{q_{\theta^{(k+1)}}(X|Y)} \cdot \frac{\pi^{(k+1)}(X)}{\pi^{(k)}(X)} \right) \right] \\
&= \mathbb{E}_{Y \sim P_Y^\star}\mathbb{E}_{X \sim q_{\theta^{(k+1)}}(\cdot|Y)} \log \frac{\pi^{(k+1)}(X)}{\pi^{(k)}(X)} - \mathbb{E}_{Y \sim P_Y^\star}\mathbb{E}_{X \sim q_{\theta^{(k+1)}}(\cdot|Y)} \log \frac{q_{\theta^{(k+1)}}(X|Y)}{P^{(k)}(X|Y)} \\
&= D_{\mathrm{KL}}(\pi^{(k+1)} \parallel \pi^{(k)}) - \mathbb{E}_{Y \sim P_Y^\star} D_{\mathrm{KL}}(q_{\theta^{(k+1)}}(\cdot|Y) \parallel P^{(k)}(\cdot|Y)).
\end{aligned}$$

Rearranging the terms completes the proof. $\qquad\square$

## B.2  PROOF OF PROPOSITION 2

We first show that: For each $k \geq 0$, it holds that

$$D_{\mathrm{KL}}(P_Y^\star \parallel P_Y^{(k+1)}) \leq D_{\mathrm{KL}}(P_X^\star \parallel \pi^{(k)}) - D_{\mathrm{KL}}(P_X^\star \parallel \pi^{(k+1)}) + \widetilde{\varepsilon}_{\mathsf{SM}}^{(k)}.$$

To simplify the presentation, we define $\widetilde{\pi}^{(k+1)}(x) = \mathbb{E}_{Y \sim P_Y^\star} P^{(k)}(x|Y)$. Then, by definition, we have

$$\begin{aligned}
\widetilde{\pi}^{(k+1)}(x) &= \mathbb{E}_{Y \sim P_Y^\star} P^{(k)}(x|Y) \\
&= \mathbb{E}_{Y \sim P_Y^\star}\left[ \frac{\pi^{(k)}(x)\mathbf{Q}(Y|x)}{P_Y^{(k)}(Y)} \right] \\
&= \pi^{(k)}(x) \cdot \mathbb{E}_{Y \sim \mathbf{Q}(\cdot|x)}\left[ \frac{P_Y^\star(Y)}{P_Y^{(k)}(Y)} \right].
\end{aligned}$$

Therefore, it follows that

$$\begin{aligned}
D_{\mathrm{KL}}(P_X^\star \parallel \pi^{(k)}) - D_{\mathrm{KL}}(P_X^\star \parallel \widetilde{\pi}^{(k+1)}) &= \mathbb{E}_{X \sim P_X^\star} \log \frac{\widetilde{\pi}^{(k+1)}(X)}{\pi^{(k)}(X)} \\
&= \mathbb{E}_{X \sim P_X^\star} \log \mathbb{E}_{Y \sim \mathbf{Q}(\cdot|x)}\left[ \frac{P_Y^\star(Y)}{P_Y^{(k)}(Y)} \right] \\
&\geq \mathbb{E}_{X \sim P_X^\star}\mathbb{E}_{Y \sim \mathbf{Q}(\cdot|x)}\left[ \log \frac{P_Y^\star(Y)}{P_Y^{(k)}(Y)} \right]
\end{aligned}$$

$$= \mathbb{E}_{Y \sim P_Y^\star} \left[ \log \frac{P_Y^\star(Y)}{P_Y^{(k)}(Y)} \right] = D_{\mathrm{KL}}(P_Y^\star \parallel P_Y^{(k)}).$$

Furthermore, we have

$$D_{\mathrm{KL}}(P_X^\star \parallel \pi^{(k+1)}) - D_{\mathrm{KL}}(P_X^\star \parallel \widetilde{\pi}^{(k+1)})$$

$$= \mathbb{E}_{X \sim P_X^\star}[\log \widetilde{\pi}^{(k+1)}(X) - \log \pi^{(k+1)}(X)]$$

$$= \mathbb{E}_{X \sim P_X^\star} \left[ \log \mathbb{E}_{Y \sim P_Y^\star}[P^{(k)}(X|Y)] - \log \mathbb{E}_{Y \sim P_Y^\star}[q_{\theta^{(k+1)}}(X|Y)] \right]$$

$$\leq \mathbb{E}_{(X,Y) \sim P_X^\star} \left[ \log \frac{P^{(k)}(X|Y)}{q_{\theta^{(k+1)}}(X|Y)} \right] = \widetilde{\varepsilon}_{\mathsf{SM}}^{(k)}.$$

Combining the above equations, we have shown that

$$D_{\mathrm{KL}}(P_Y^\star \parallel P_Y^{(k)}) \leq D_{\mathrm{KL}}(P_X^\star \parallel \pi^{(k)}) - D_{\mathrm{KL}}(P_X^\star \parallel \pi^{(k+1)}) + \widetilde{\varepsilon}_{\mathsf{SM}}^{(k)}.$$

This is the desired upper bound. Taking the summation over $k = 0, 1, \cdots, K$ completes the proof. For the last-iterate convergence rate, we only need to use the fact that $D_{\mathrm{KL}}(P_Y^\star \parallel P_Y^{(k)}) \leq D_{\mathrm{KL}}(P_Y^\star \parallel P_Y^{(K)}) + \sum_{\ell=k}^{K} \varepsilon_{\mathsf{SM}}^{(\ell)}$ (by Lemma 1). $\qquad \square$

### B.3 PROOF OF PROPOSITION 3

By Proposition 2, we have

$$D_{\mathrm{KL}}(P_X^\star \parallel \pi^{(k+1)}) + D_{\mathrm{KL}}(P_Y^\star \parallel P_Y^{(k+1)}) \leq D_{\mathrm{KL}}(P_X^\star \parallel \pi^{(k)}) + \widetilde{\varepsilon}_{\mathsf{SM}}^{(k)}.$$

Using Assumption 1, we know that as long as $D_{\mathrm{KL}}(P_X^\star \parallel \pi^{(k)}) \leq R$, we have

$$(1 + \kappa^{-1}) D_{\mathrm{KL}}(P_X^\star \parallel \pi^{(k+1)}) \leq D_{\mathrm{KL}}(P_X^\star \parallel \pi^{(k)}) + \widetilde{\varepsilon}_{\mathsf{SM}}^{(k)}.$$

Denote $\widetilde{\varepsilon}_{\mathsf{SM}} = \max_k \widetilde{\varepsilon}_{\mathsf{SM}}^{(k)}$. Therefore, using the fact that $\widetilde{\varepsilon}_{\mathsf{SM}}^{(k)} \leq \widetilde{\varepsilon}_{\mathsf{SM}} \leq \frac{R}{\kappa}$, we can show by induction that $D_{\mathrm{KL}}(P_X^\star \parallel \pi^{(k)}) \leq R$ for each $k \geq 0$, and hence

$$(1 + \kappa^{-1}) D_{\mathrm{KL}}(P_X^\star \parallel \pi^{(k+1)}) \leq D_{\mathrm{KL}}(P_X^\star \parallel \pi^{(k)}) + \widetilde{\varepsilon}_{\mathsf{SM}}.$$

Applying this inequality recursively, we obtain

$$D_{\mathrm{KL}}(P_X^\star \parallel \pi^{(k)}) \leq \frac{\kappa}{1 + \kappa} D_{\mathrm{KL}}(P_X^\star \parallel \pi^{(k-1)}) + \widetilde{\varepsilon}_{\mathsf{SM}}$$

$$\leq \left( \frac{\kappa}{1 + \kappa} \right)^2 D_{\mathrm{KL}}(P_X^\star \parallel \pi^{(k-2)}) + \left( \frac{\kappa}{1 + \kappa} \right) \widetilde{\varepsilon}_{\mathsf{SM}} + \widetilde{\varepsilon}_{\mathsf{SM}}$$

$$\leq \cdots$$

$$\leq \left( \frac{\kappa}{1 + \kappa} \right)^k D_{\mathrm{KL}}(P_X^\star \parallel \pi^{(0)}) + \sum_{i=0}^{k-1} \left( \frac{\kappa}{1 + \kappa} \right)^{k-1-i} \widetilde{\varepsilon}_{\mathsf{SM}}$$

$$\leq e^{-k/(\kappa+1)} D_{\mathrm{KL}}(P_X^\star \parallel \pi^{(0)}) + (1 + \kappa) \widetilde{\varepsilon}_{\mathsf{SM}},$$

where the last inequality follows from $\frac{\kappa}{1+\kappa} = 1 - \frac{1}{1+\kappa} \leq \exp\left( -\frac{1}{1+\kappa} \right)$. $\qquad \square$

### B.4 RELATION BETWEEN THE SCORE-MATCHING ERRORS

In this section, we provide the following upper bound for $\widetilde{\varepsilon}_{\mathsf{SM}}^{(k)}$ in terms of $\varepsilon_{\mathsf{SM}}^{(k)}$. Recall that $\widetilde{\varepsilon}_{\mathsf{SM}}^{(k)}$ is defined as

$$\widetilde{\varepsilon}_{\mathsf{SM}}^{(k)} = \mathbb{E}_{(X,Y) \sim P^\star} \left[ \log \frac{P^{(k)}(X|Y)}{q_{\theta^{(k+1)}}(X|Y)} \right],$$

**Proposition 4.** *Suppose that $\mathbb{E}_{Y \sim P_Y^\star} D_{\chi^2}(P^\star(\cdot|Y) \parallel q_{\theta^{(k+1)}}(\cdot|Y)) \leq C < +\infty$. Then it holds that*
$$\widetilde{\varepsilon}_{\mathsf{SM}}^{(k)} \leq 2\sqrt{(C+1)\varepsilon_{\mathsf{SM}}^{(k)}}.$$

*Proof of [Proposition 4](#).* By definition,

$$
\widetilde{\varepsilon}_{\mathsf{SM}}^{(k)} \leq \mathbb{E}_{(X,Y)\sim P_X^\star} \left( \log \frac{P^{(k)}(X|Y)}{q_{\theta^{(k+1)}}(X|Y)} \right)_+
$$

$$
= \mathbb{E}_{Y\sim P_Y^\star} \mathbb{E}_{X\sim P_X^\star(\cdot|Y)} \left( \log \frac{P^{(k)}(X|Y)}{q_{\theta^{(k+1)}}(X|Y)} \right)_+
$$

$$
\leq \mathbb{E}_{Y\sim P_Y^\star} \sqrt{ \left( 1 + D_{\chi^2}(P^\star(\cdot|Y) \,\|\, q_{\theta^{(k+1)}}(\cdot|Y)) \right) \cdot \mathbb{E}_{X\sim q_{\theta^{(k+1)}}(\cdot|Y)} \left( \log \frac{P^{(k)}(X|Y)}{q_{\theta^{(k+1)}}(X|Y)} \right)_+^2 }
$$

$$
\leq \mathbb{E}_{Y\sim P_Y^\star} \sqrt{ \left( 1 + D_{\chi^2}(P^\star(\cdot|Y) \,\|\, q_{\theta^{(k+1)}}(\cdot|Y)) \right) \cdot 4 D_{\mathrm{KL}}(q_{\theta^{(k+1)}}(\cdot|Y) \,\|\, P^{(k)}(\cdot|Y)) }
$$

$$
\leq 2\sqrt{(C+1)\varepsilon_{\mathsf{SM}}^{(k)}},
$$

where we apply [Lemma 5](#). This yields the desired upper bound. $\qquad\square$

**Lemma 5.** *For any distributions $P$ and $Q$, it holds that*

$$
\mathbb{E}_{X\sim Q}(\log P(X) - \log Q(X))_+^2 \leq 4 D_{\mathrm{KL}}(Q \,\|\, P).
$$

*Proof.* Note that $\log x \leq 2(\sqrt{x}-1)$ for any $x \geq 1$, and hence $(\log x)_+^2 \leq 4(\sqrt{x}-1)^2$. Applying this inequality, we have

$$
\mathbb{E}_{X\sim Q}(\log P(X) - \log Q(X))_+^2 = \mathbb{E}_{X\sim Q}\left( \log \frac{P(X)}{Q(X)} \right)_+^2
$$

$$
\leq 4\mathbb{E}_{X\sim Q}\left( \sqrt{\frac{P(X)}{Q(X)}} - 1 \right)^2 = 8 D_{\mathrm{H}}^2(P,Q) \leq 4 D_{\mathrm{KL}}(Q \,\|\, P).
$$

This is the desired upper bound. $\qquad\square$

## C    EXPERIMENT DETAILS

**Parametrization.**    Following Section 2.2, we adopt the denoiser parametrization $d_\theta(x, t|y)$, and the conditional score function $\mathbf{s}_\theta$ is thus given by

$$\mathbf{s}_\theta(x, t|y) = \frac{d_\theta(x, t|y) - x}{\sigma_t^2}.$$

Therefore, the score-matching loss defined in (10) can be equivalently written as

$$L_{\text{SM},k}(\theta) = \int_0^1 \lambda_t' \mathbb{E}_{X_0 \sim \mathcal{D}^{(k)}, Y \sim \mathbf{Q}(\cdot|X)} \mathbb{E}_{X_t \sim \mathsf{N}(X_0, \sigma_t^2 \mathbf{I})} \left\| d_\theta(X_t, t|Y) - X_0 \right\|^2 dt, \qquad (12)$$

where $\lambda_t' = \frac{\lambda_t}{\sigma_t^2}$, and $\lambda_t$ is the weight function from (10).

In our experiments, we adopt the following noise schedule:

$$\sigma_t^2 = \exp\left((1 - t)\log(\sigma_0) + t\log(\sigma_1)\right),$$

where $\sigma_0 < \sigma_1$ are appropriate parameters, and the scalar $\sigma_t$ is encoded as a vector embedding. The input to the denoiser network is the concatenation of $X_t$, $Y$, and the vector embedding of the noise $\sigma_t$. We also choose $\lambda_t = (\sigma_t^2 + 1) \cdot f(t; \alpha, \beta)$, where $f(t; \alpha, \beta)$ is the density function of the Beta distribution with parameters $(\alpha, \beta)$.

For the manifold experiment (Appendix C.2), we choose $\alpha = 3.5, \beta = 1.5, \sigma_0 = 10^{-3}, \sigma_1 = 10^1$. For the remaining experiments, we set $\alpha = \beta = 3, \sigma_0 = 10^{-3}, \sigma_1 = 10^2$.

**Initialization.**    As noted in Section 3, the convergence rate of DiffEM depends on the quality of the initial prior $\pi^{(0)}$ through the quantity $D_{\text{KL}}(P_X^\star \| \pi^{(0)})$, i.e., the KL divergence between the ground-truth data distribution $P_X^\star$ and the initial $\pi^{(0)}$. Therefore, a better initial prior may lead to faster convergence. In our experiments, we consider the following initialization strategies:

(a) **Corrupted prior:** For (2), the observation is $Y = (AX + \epsilon, A)$. When $d_y = d_x$, we can consider the *corrupted prior* $\pi^{(0)}$, which is simply the distribution of $X' = AX + \epsilon$. To sample from $\pi^{(0)}$, we can draw $Y = (AX + \epsilon, A) \sim P_Y^\star$ and set $X' = Y[0 : d_y]$.

(b) **Gaussian prior:** In general, we can fit a Gaussian prior $\pi^{(0)} = \mathsf{N}(\mu_X, \Sigma_X)$ using the observations $\{Y^{[1]}, \cdots, Y^{[N]}\} \sim P_Y^\star$.

(c) **Warm-start:** More generally, we can set $\pi^{(0)}$ to be any pre-trained diffusion prior as the *warm-start* prior. In particular, this can be the diffusion prior trained on corrupted data by existing methods (Daras et al., 2023b; Kawar et al., 2023; Rozet et al., 2024, etc.).

For the experiments (except Section 4.1.1), we adopt initialization strategy (b). Following the implementation in (Rozet et al., 2024), the Gaussian prior is fitted efficiently through a few closed-form EM iterations. An exception is the experiment on blurred CIFAR-10, where we adopt strategy (a). In Section 4.1.1, we perform experiments with strategy (c), applying DiffEM to the warm-start prior trained by EM-MMPS (Rozet et al., 2024), demonstrating that DiffEM can monotonically improve upon the initial prior.

### C.1    ADDITIONAL EXPERIMENT: SYNTHETIC MANIFOLD IN $\mathbb{R}^5$

We evaluate our method's performance on a synthetic problem introduced by (Rozet et al., 2024). In this setting, the latent space is $\mathcal{X} = \mathbb{R}^5$, with the latent distribution $P_X^\star$ supported on a one-dimensional curve in $\mathbb{R}^5$. The observation $Y = (AX + \epsilon, A)$ is generated through the following steps: (1) sample a latent point $X \sim P_X^\star$, (2) sample a corruption matrix $A \in \mathbb{R}^{2 \times 5} \sim P_A$ with rows drawn uniformly from the unit sphere $\mathbb{S}^4$, and (3) add Gaussian noise $\epsilon \sim \mathsf{N}\left(0, \sigma_Y^2 \mathbf{I}\right)$.

Following Rozet et al. (2024), we apply our method to a dataset of 65536 independent observations with noise variance $\sigma_Y^2 = 10^{-4}$. Detailed experimental settings are presented in Appendix C.2. Figure 3 illustrates the two-dimensional marginals of the reconstructed latent distribution compared to those obtained by (Rozet et al., 2024). The results demonstrate that our method achieves better concentration around the ground-truth curve, providing empirical evidence that the conditional diffusion model learns the posterior distribution more accurately than the approximate posterior sampling scheme of (Rozet et al., 2024) (cf. Section 2.1).

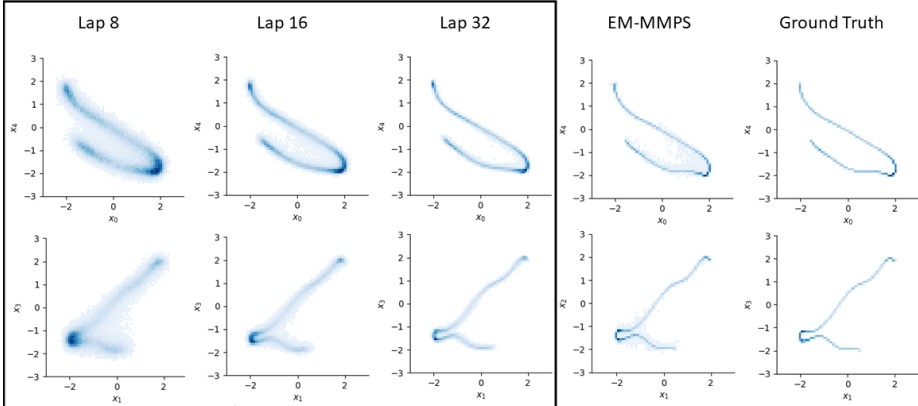

Figure 3: Evolution of the learned latent distribution on the synthetic manifold task. From left to right: reconstructed distributions from our model at DiffEM iterations 8, 16, and 32, followed by the distribution from EM-MMPS ((Rozet et al., 2024), 32th iteration) and the ground-truth $P_X^\star$. Our method shows progressively better concentration around the ground-truth curve, demonstrating more accurate posterior learning compared to previous work.

## C.2   MORE DETAILS ON THE EXPERIMENT IN APPENDIX C.1

We implement the denoiser network $d_\theta(x, t|y)$ using a Multi-Layer Perceptron (MLP). The network architecture and training hyperparameters are detailed in Table 5.

| | |
|---|---|
| Architecture | MLP |
| Input Shape | $5 + 2 + 5 \times 2 = 17$ |
| Hidden Layers | 3 |
| Hidden Layer Sizes | 256, 256, 256 |
| Activation | SiLU |
| Normalization | LayerNorm |
| Optimizer | Adam |
| Weight Decay | 0 |
| Scheduler | linear |
| Initial Learning Rate | $1 \times 10^{-3}$ |
| Final Learning Rate | $1 \times 10^{-6}$ |
| Gradient Norm Clipping | 1.0 |
| Batch Size | 1024 |
| Epochs in each iteration | 65536 |
| Sampler | Predictor-Corrector |
| Sampler Steps | 4096 |
| Number of EM iterations | 32 |

Table 5: Network architecture and training hyperparameters for the MLP used in the synthetic manifold experiment.

To quantify the quality of the learned distribution, we compute the Sinkhorn divergence $S_\lambda$ Ramdas et al. (2015) with regularization parameter $\lambda = 10^{-3}$ after each epoch. The Sinkhorn divergence is defined as:

$$S_\lambda(\mu, \nu) := T_\lambda(\mu, \nu) - \frac{1}{2}(T_\lambda(\mu, \mu) + T_\lambda(\nu, \nu))$$

$$T_\lambda(\mu, \nu) := \min_{\gamma \in \Pi(\mu, \nu)} \int_{(\mathbb{R}^d)^2} \|y - x\|_2^2 d\gamma(x, y) + 2\lambda H(\gamma, \mu \otimes \nu)$$

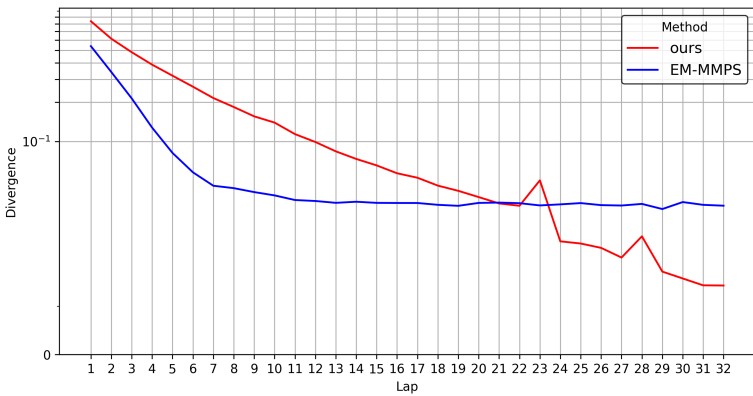

Figure 4: Evolution of Sinkhorn divergence between the ground-truth and reconstructed distributions during training. The red line shows DiffEM, and the blue line shows EM-MMPS.

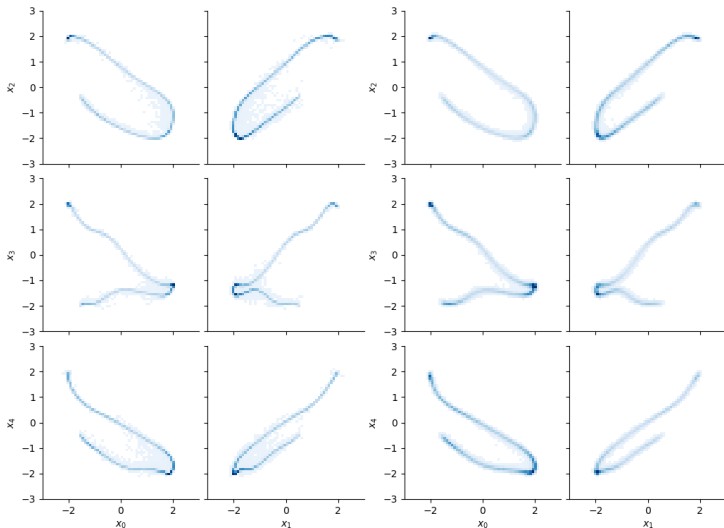

Figure 5: Comparison of 2D marginals of reconstructed distributions after the final iteration. Left: EM-MMPS; Right: DiffEM. DiffEM achieves better concentration around the ground-truth curve, indicating more accurate posterior learning.

We plot the evolution of Sinkhorn divergence over the iterations of DiffEM and EM-MMPS (Rozet et al., 2024) in Figure 4. We also plot the 2D marginals of the distributions reconstructed by DiffEM and EM-MMPS in Figure 5.

Figure 4 demonstrates that while EM-MMPS provides effective initialization when the learned distribution is far from the true data distribution, it plateaus quickly and fails to achieve further improvements. This is likely due to the inherent approximation error of the approximate posterior sampling scheme (MMPS). In contrast, DiffEM continues to refine the reconstructed distribution, achieving better concentration around the ground-truth curve.

### C.3  DETAILS OF MASKED CIFAR-10 (SECTION 4.1)

In this experiment, the conditional denoiser network $d_\theta$ is a U-Net Ronneberger et al. (2015), and we adopt the same experimental setup as Rozet et al. (2024) for a fair comparison. The only major difference in the architecture arises from the fact that our model is conditional and thus for the input we need to feed two images $X_t$ with shape $(32, 32, 3)$ and $Y$ with shape $(32, 32, 3)$ to the model, we concatenate the images on the third dimension and thus the input shape for the model is $(32, 32, 6)$,

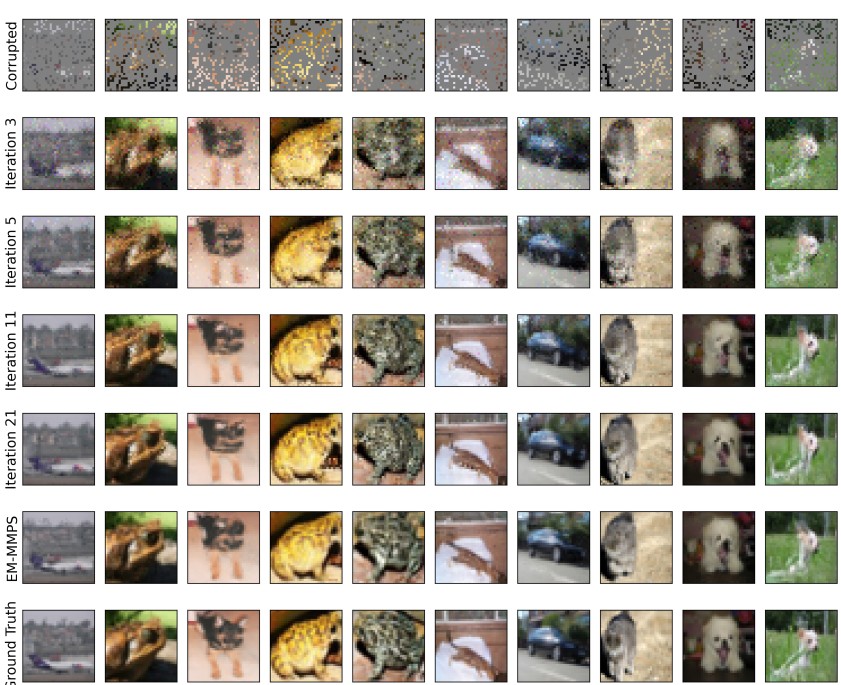

Figure 6: Qualitative comparison of reconstruction results on masked CIFAR-10 images. Top to bottom: corrupted input, EM-MMPS reconstructions, DiffEM reconstructions, and ground truth.

the output is also $(32, 32, 6)$ but in the whole training process we ignore the last three channels of the output. The details of network architecture and hyperparameters are presented in Table 6.

| Experiment | CIFAR-10 | CelebA |
|---|---|---|
| Architecture | U-Net | U-Net |
| Input Shape | (32, 32, 6) | (64, 64, 6) |
| Channels Per Level | (128, 256, 384) | (128, 256, 384, 512) |
| Attention Heads per level | (0, 4, 0) | (0, 0, 0, 4) |
| Hidden Blocks | (5, 5, 5) | (3, 3, 3, 3) |
| Kernel Shape | (3, 3) | (3, 3) |
| Embedded Features | 256 | 256 |
| Activation | SiLU | SiLU |
| Normalization | LayerNorm | LayerNorm |
| Optimizer | Adam | Adam |
| Initial Learning Rate | $2 \times 10^{-4}$ | $1 \times 10^{-4}$ |
| Final Leanring Rate | $1 \times 10^{-6}$ | $1 \times 10^{-6}$ |
| Weight Decay | 0 | 0 |
| EMA | 0.9999 | 0.999 |
| Dropout | 0.1 | 0.1 |
| Gradient Norm Clipping | 1.0 | 1.0 |
| Batch Size | 256 | 256 |
| Epochs per EM iteration | 256 | 64 |
| Sampler | DDPM | DDPM |

Table 6: Network architecture and training hyperparameters for the U-Net models used in the CIFAR-10 and CelebA experiments. Input shape varies by task.

We apply DiffEM with $K = 21$ iterations to train our conditional diffusion model and evaluate its performance for the posterior sampling task as described in Section 4.1. To evaluate the quality of the reconstructed data distribution, we also train an unconditional diffusion model with the same

architecture on the reconstructed data. We compute the Inception Score (IS) Salimans et al. (2016) and the Fréchet Inception Distance (FID) Heusel et al. (2017) using the torch-fidelity package (Obukhov et al., 2021), and $FD_{DINOv2}$ (Oquab et al., 2023; Stein et al., 2023) and $FD_\infty$ (Chong and Forsyth, 2020) using the codebase from (Stein et al., 2023). The results are presented in Table 1 and Table 2. We also note that the results of EM-MMPS are obtained with 32 iterations, following the original setup of Rozet et al. (2024).

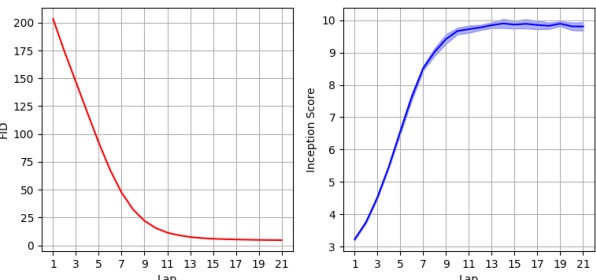

Figure 7: Evolution of evaluation metrics for posterior sampling measured during DiffEM training on CIFAR-10 with random masking. Left: FID, Right: Inception Score.

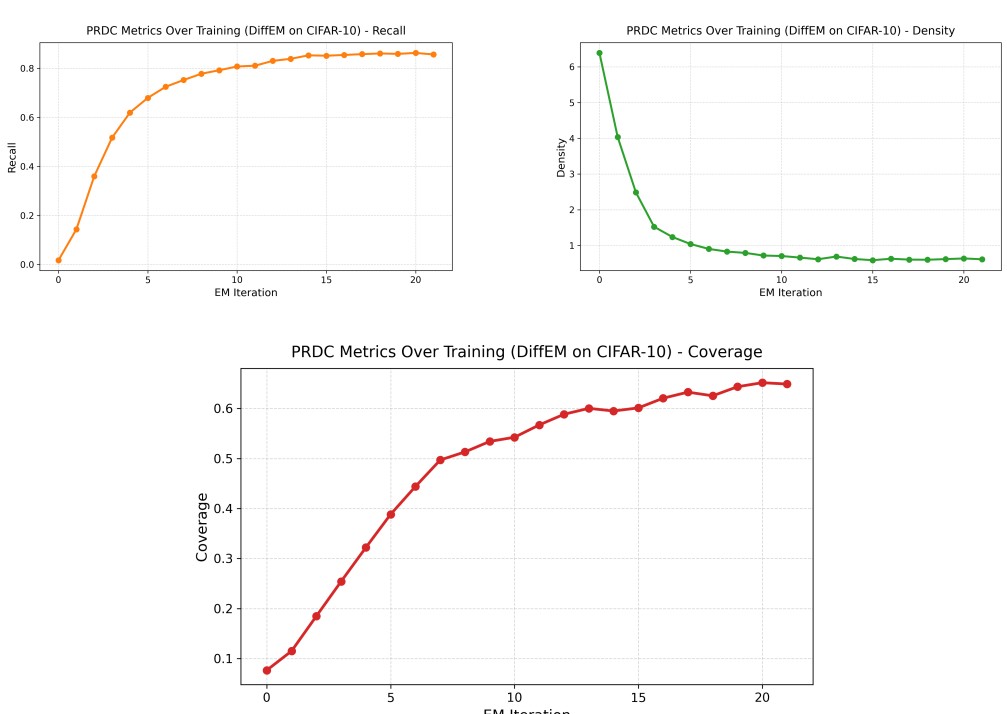

Figure 8: Evolution of evaluation metrics for posterior sampling measured during DiffEM training on CIFAR-10 with random masking.

As an illustration, we also plot the evolution of the IS and FID during DiffEM iterations, demonstrating that DiffEM monotonically improves the quality of the reconstructed data distribution, in accordance with our theoretical results (Lemma 1).

**Experiments with higher corruption.** In addition, we perform experiments on CIFAR-10 with corruption probability $\rho = 0.9$ (i.e., 90% of the pixels are randomly deleted) and present the results in Table 7. Under such high corruptions, DiffEM also consistently outperforms EM-MMPS (Rozet et al., 2024).

| Task | Method | IS ↑ | FID ↓ | FD$_{\text{DINOv2}}$ ↓ | FD$_\infty$ ↓ |
|------|--------|------|-------|-------------------------|----------------|
| Posterior sampling | EM-MMPS | 5.06 | 67.97 | 1045.51 | 1039.82 |
|  | DiffEM | **5.86** | **46.13** | **915.69** | **912.26** |
| Unconditional generation | EM-MMPS | 4.86 | 73.34 | 1174.13 | 1168.66 |
|  | DiffEM | **5.46** | **49.10** | **1111.16** | **1107.64** |

Table 7: Performance of DiffEM and EM-MMPS on CIFAR-10 with $90\%$ random masking.

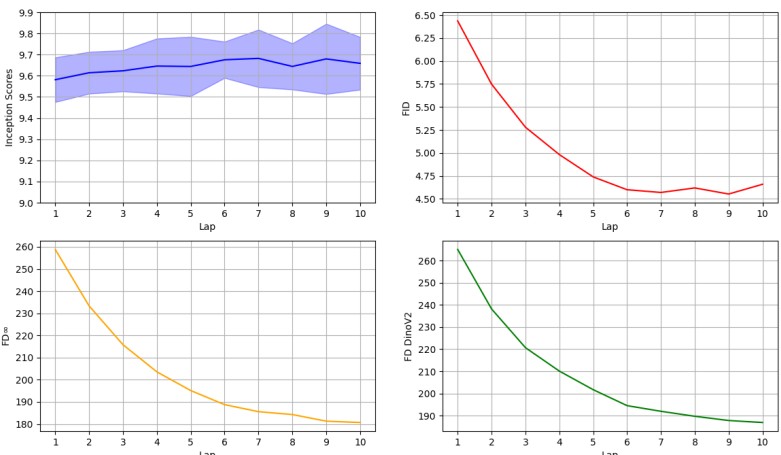

Figure 9: Evolution of IS, FID, DINO, FD$_\infty$ during the 10 DiffEM iterations with the warm-started prior.

### C.4 DIFFEM WITH WARM-START

We plot the evolution of IS, FID, FD$_{\text{DINOv2}}$ and FD$_\infty$ scores during training in Figure 9.

### C.5 DETAILS OF BLURRED CIFAR-10

In the experiment on CIFAR-10 with Gaussian blur, we set $\sigma_{\text{kernel}} = 2$ and $\sigma_Y^2 = 10^{-6}$. We apply DiffEM for $K = 21$ iterations, with the same initialization, denoiser network architecture, and hyperparameters as in the masked CIFAR-10 experiment (detailed in Table 6, Appendix C.3). Due to time constraints, we do not evaluate EM-MMPS (Rozet et al., 2024), as the moment-matching steps (based on the conjugate gradient method) are very time-consuming in this setting.

**Qualitative study.** To evaluate the quality of the trained conditional model, we sample a set of blurred images from the CIFAR-10 training set and use the trained model to generate a reconstruction for each image. We present the images in Figure 10.

**Quantitative comparison.** For comparison, we use the Richardson-Lucy deblurring algorithm Richardson (1972) as a baseline, which is a widely used method for image deconvolution. We also plot the evolution of the IS and FID during DiffEM iterations in Figure 11.

| Method | IS ↑ | FID ↓ | FD$_{\text{DINOv2}}$ ↓ | FD$_\infty$ ↓ |
|--------|------|-------|-------------------------|----------------|
| Richardson-Lucy deconvolution | 3.72 | 131.74 | 1479.79 | 1470.78 |
| DiffEM (Ours) | **6.12** | **43.65** | **404.05** | **400.65** |

Table 8: Posterior sampling performance on CIFAR-10 with Gaussian blur ($\sigma_{\text{kernel}} = 2$).

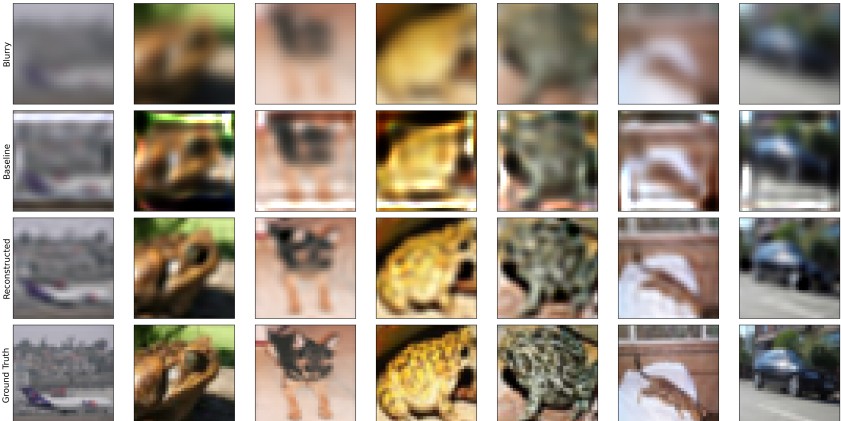

Figure 10: Qualitative results of image reconstruction from Gaussian blur. Top to bottom: blurred image, reconstruction by Richardson-Lucy deconvolution, image reconstructed by DiffEM model, and ground truth. DiffEM effectively recovers image details.

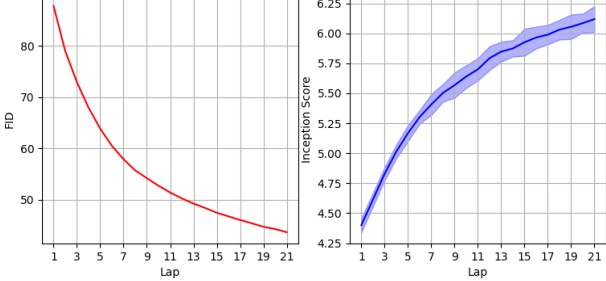

Figure 11: Evolution of evaluation metrics for posterior sampling measured during DiffEM training on CIFAR-10 with Gaussian blur. Left: FID, Right: Inception Score.

| Method | IS ↑ | FID ↓ | FD$_{\text{DINOv2}}$ ↓ | FD$_\infty$ ↓ |
|---|---|---|---|---|
| DiffEM (Ours) | 11.27 | 51.25 | 772.23 | 768.19 |

Table 9: unconditional generation performance on CIFAR-10 with Gaussian blur ($\sigma_{\text{kernel}} = 2$).

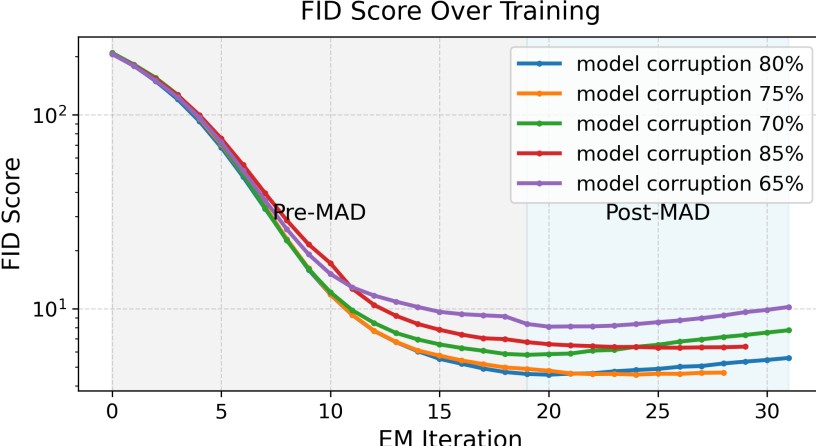

Figure 12: FID over different EM iterations are shown for each of the corruptions assumed by the model, details of the experiment are discussed in C.6

## C.6 CORRUPTION MODEL MISMATCH

In many real-world settings, the likelihood function is not known exactly. Instead, one typically works with an estimate $\hat{Q}(\cdot \mid X)$ rather than the true likelihood function $Q(\cdot \mid X)$. Notably, all of our experiments and those in prior work (Rozet et al., 2024; Daras et al., 2023b; 2025b), assume access to the exact likelihood function.

In this section, we investigate the more realistic scenario in which the data are corrupted by one channel while the model is trained using a misspecified one. Concretely, we use CIFAR-10 and apply random masking with true corruption probability $\rho = 0.75$ to generate the observations. However, during training we assume a mismatched corruption probability $\hat{\rho} = \rho + \Delta$. For $\Delta \in \{-0.1, -0.05, 0, 0.05, 0.1\}$, we train and evaluate DiffEM to study its robustness under corruption-model misspecification. Based on the results shown Figure 12, slightly over estimating the corruption probability, which will make the model be trained on a harder task would yield a better result than slightly underestimating the corruption probability.

## C.7 NON-LINEAR DISCRETE CORRUPTION

In this section, we investigate a corruption function that is neither linear nor continuous, but instead exhibits inherently discrete behavior. A canonical example of such corruption is JPEG compression. JPEG applies a sequence of nonlinear operations—including blockwise discrete cosine transforms (DCT), quantization, and rounding—which introduce structured, non-Gaussian artifacts that cannot be modeled as additive noise. This setting is especially relevant, as many real-world image pipelines (e.g., internet images, mobile devices, and storage-limited datasets) rely heavily on JPEG or similar codecs.

To study the effect of such discrete corruption, we compress and decompress all CIFAR-10 images using JPEG with a quality factor of $0.2$. At this low quality level, the quantization step is extremely aggressive, removing a substantial portion of the high-frequency content and producing severe compression artifacts. In practice, this corruption destroys a significant amount of the original information in the dataset. We train our diffusion models directly on these JPEG-compressed images to evaluate the robustness of our method under realistic, non-smooth likelihood functions that differ significantly from the Gaussian processes assumed in prior work.

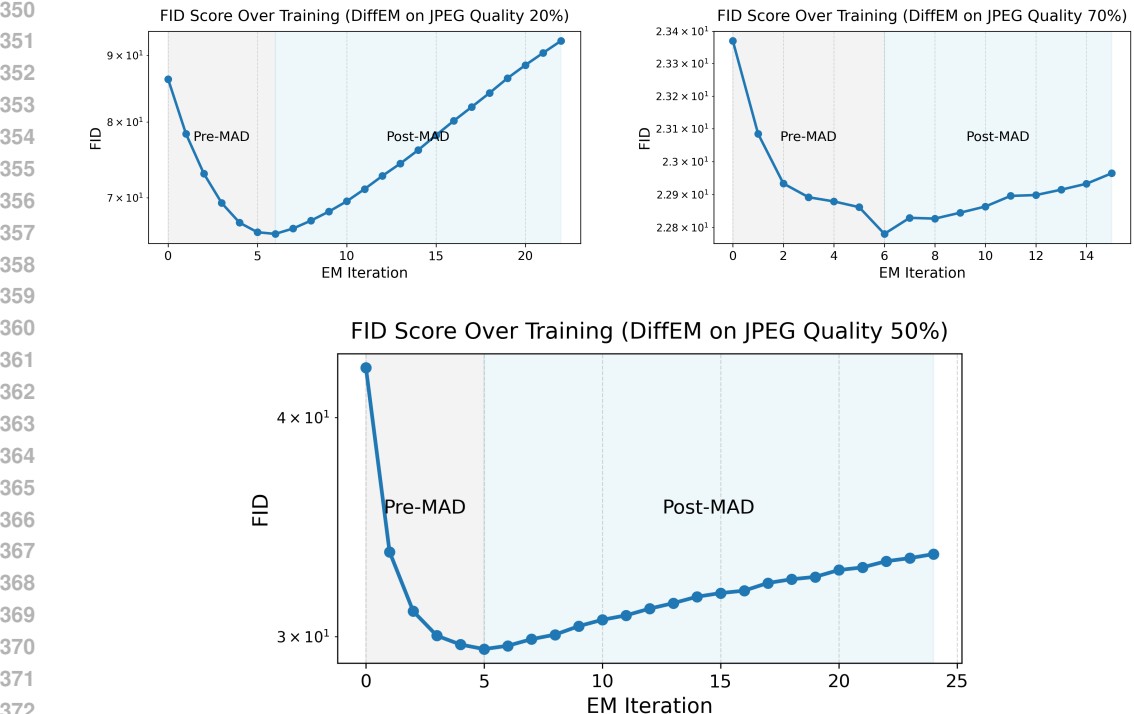

Figure 13: Evolution of the FID for DiffEM's conditional model under JPEG corruption with quality 20% on CIFAR-10. More details on the experiment are provided in Section C.7.

The results in Figure 13 indicate that under this high level of corruption the model converges rapidly and exhibits the MAD (Manifold Attracted Diffusion) effect much earlier than in our other experiments. Further discussion of MAD is provided in Section C.8. Notably, this experiment also shows that MAD can have a pronounced impact: after sufficient EM iterations, performance may degrade significantly, with the model at iteration 21 performing worse (in terms of distributional metrics) than after a single iteration.

## C.8 MAD: Manifold Attracted Diffusion

We consistently observe the MAD effect (Elbrächter et al., 2025) across nearly all of our experiments when the EM procedure is continued for sufficiently many iterations. In the case of CIFAR-10 with random masking at corruption level $\rho = 0.75$, evaluating the conditional model after each EM iteration yields the behavior shown in Figure 14. Figure Figure 13 demonstrates the MAD effect for JPEG corruption under three different compression qualities (20%, 50%, and 70%). The results suggest that stronger corruption leads to more pronounced MAD behavior. An interesting phenomenon emerges when we relate this observation to the theoretical bound in Proposition 3. In that bound, the second term is non-decreasing in $K$, while the first term becomes negligible for sufficiently large $K$, as it decays exponentially fast. Under the proposition's assumption

$$\varepsilon_x^{(k)} \leq \frac{R}{\kappa},$$

we obtain the bound

$$D_{\mathrm{KL}}(P_X^\star \parallel \pi^{(K)}) \leq \exp\left(-\frac{K}{\kappa+1}\right) D_{\mathrm{KL}}(P_X^\star \parallel \pi^{(0)}) + \frac{\kappa+1}{\kappa} R.$$

Because $\kappa \geq 1$ (as guaranteed by the identifiability condition in Assumption 1), we have $\frac{\kappa+1}{\kappa} R \leq 2R$. Thus, after sufficiently many EM iterations, the model cannot drift arbitrarily far from the true distribution: once it reaches a distance larger than $2R$, it is forced to move back with an exponential rate. Indeed, Figure 13 shows that under *reasonable* corruption levels, the MAD effect is always present but remains well-controlled.

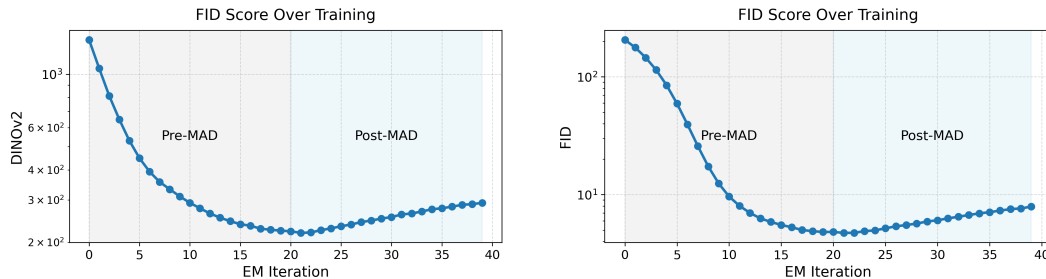

Figure 14: Evolution of DINOv2 and Fréchet Inception Distance across EM iterations for conditional model. After an initial phase of improvement, both metrics begin to gracefully degrade over later iterations. Experiment was done on CIFAR-10 with $\rho = 0.75$ and discretization step $128$.

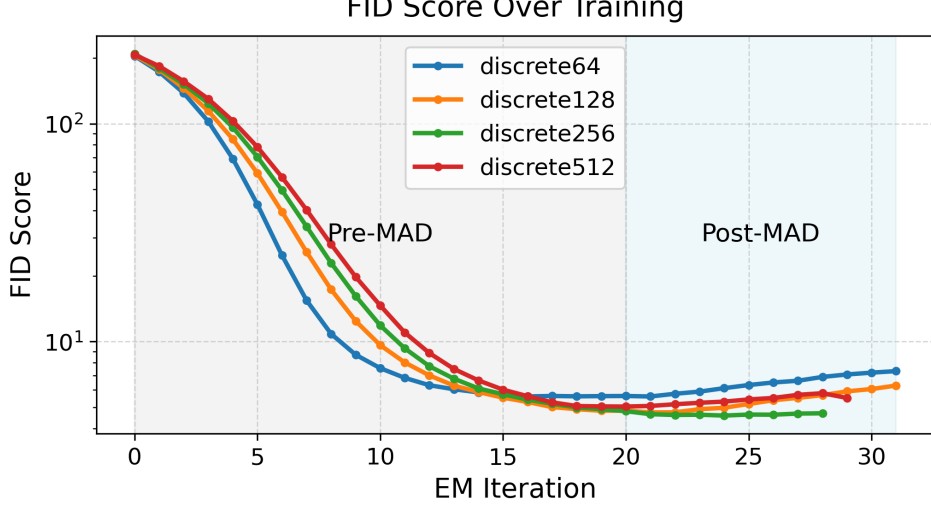

Figure 15: Evolution of the conditional model's performance across EM iterations under different discretization choices for the CIFAR-10 masking experiment with $\rho = 0.75$.

However, when the corruption becomes very severe (e.g., JPEG quality $20\%$), the model can diverge significantly after many EM iterations. This is expected, because in such regimes the identifiability assumption no longer holds due to substantial information loss introduced by the corruption channel. In summary, the MAD effect always appears, but under moderate corruption—where identifiability is valid—it remains controlled. For high corruption levels, where the corruption channel destroys too much information, no theoretical guarantees prevent the MAD effect from becoming extreme.

### C.9 ANALYSIS OF DISCRETIZATION ERROR

In Section 3, we decomposed the score-matching error $\varepsilon_{\mathrm{KL}}^{(k)}$ into a discretization error and a learning error. In this section, we examine how the model's performance varies under different discretization choices. We train on randomly masked CIFAR-10 with corruption probability $\rho = 0.75$, and for discretization step counts $N \in \{64, 128, 256, 512\}$ we train the model and evaluate it after each EM iteration. The resulting performance curves are shown in Figure 15.

We observe that in the beginning iterations there is a large performance gap and then it closes to a smaller gap. We decomposed the score-matching error $\varepsilon_{\mathrm{KL}}^{(k)}$ into a discretization error and a learning error. In the beginning iterations the learning error is high and is causing the large gaps and when the learning error is decreased we could see the discretization error causing the gaps in performances.

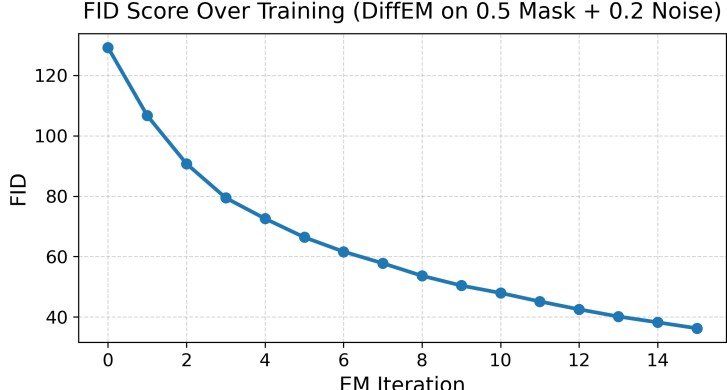

Figure 16: Conditional model's FID evolution over EM iterations for experiment done in C.10. The dataset is corrupted by sampling $A(X + \sigma N)$ where $A_{ij} \sim \text{Ber}(\frac{1}{2})$, $N \sim \mathcal{N}(0, I)$ and $\sigma = 0.2$. Images are normalized so that $X_i \in [-2, 2]$.

### C.10 MASKING + GAUSSIAN NOISE CORRUPTION

We also evaluate our method under a mixed corruption model combining masking with additive Gaussian noise. Specifically, we run the CIFAR-10 experiment with a masking probability of $\rho = 0.5$, which is milder than the high-corruption setting $\rho = 0.75$, and additionally add Gaussian noise with standard deviation $\sigma = 0.2$. The resulting likelihood function is

$$Q(Y \mid X) = A(X + \sigma Z), \quad Z \sim \mathcal{N}(0, I), \quad A_{ij} \sim \text{Ber}(0.75),$$

where $A$ denotes the random masking matrix. Qualitative samples are shown in Figure 17, and the evolution of evaluation metrics across EM iterations is presented in Figure 16.

### C.11 MASKED CELEBA

As a demonstration, we sample seven masked images from the CelebA training set under the $75\%$ corruption setting. Using the trained model, we generate reconstructions for each image after the 1st, 2nd, 4th, 8th, and 16th iterations. The results are shown in Appendix C.11.

The denoiser architecture is detailed in Table 6. For the $50\%$ corruption setting, we trained the conditional diffusion model for 20 EM iterations, while for the $75\%$ corruption setting we trained it for 24 iterations. In both cases, we trained EM-MMPS for 9 iterations. The computational overhead of Moment Matching Posterior Sampling becomes particularly evident in this experiment, as the CelebA dataset is larger (202,599 images) and each image is higher-dimensional ($64 \times 64$) compared to CIFAR-10. We observed that each EM iteration of EM-MMPS required $4.85 \pm 0.02$ hours, whereas each iteration of DiffEM required $1.19 \pm 0.03$ hours.

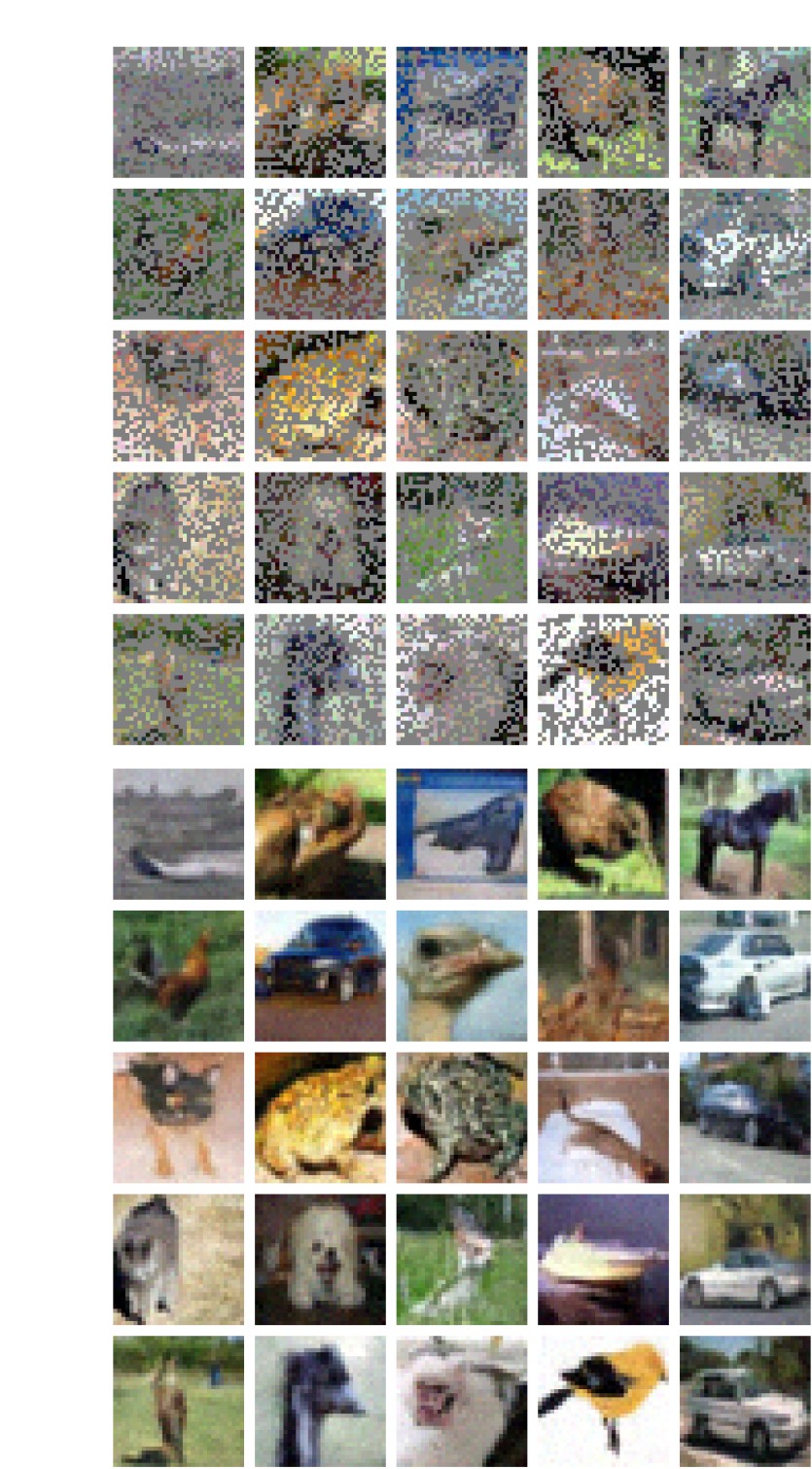

Figure 17: Samples from the Experiment in section C.10, top half showing the samples in the corrupted dataset that the model has access to and the bottome half shows the samples generated by the conditional model.

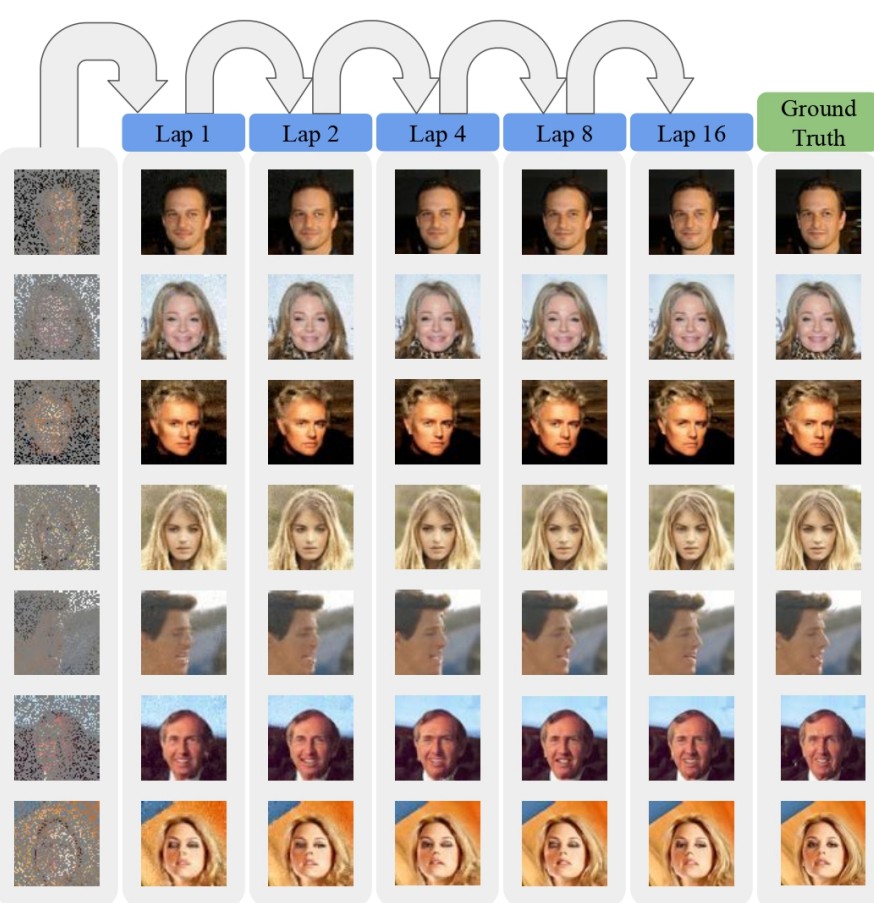

Figure 18: Qualitative results on the CelebA experiment under the $75\%$ corruption setting. The leftmost column shows samples from the dataset. The subsequent columns display reconstructions generated by the conditional diffusion model after laps $k = 1, 2, 4, 8, 16$. The rightmost column shows the ground-truth images.

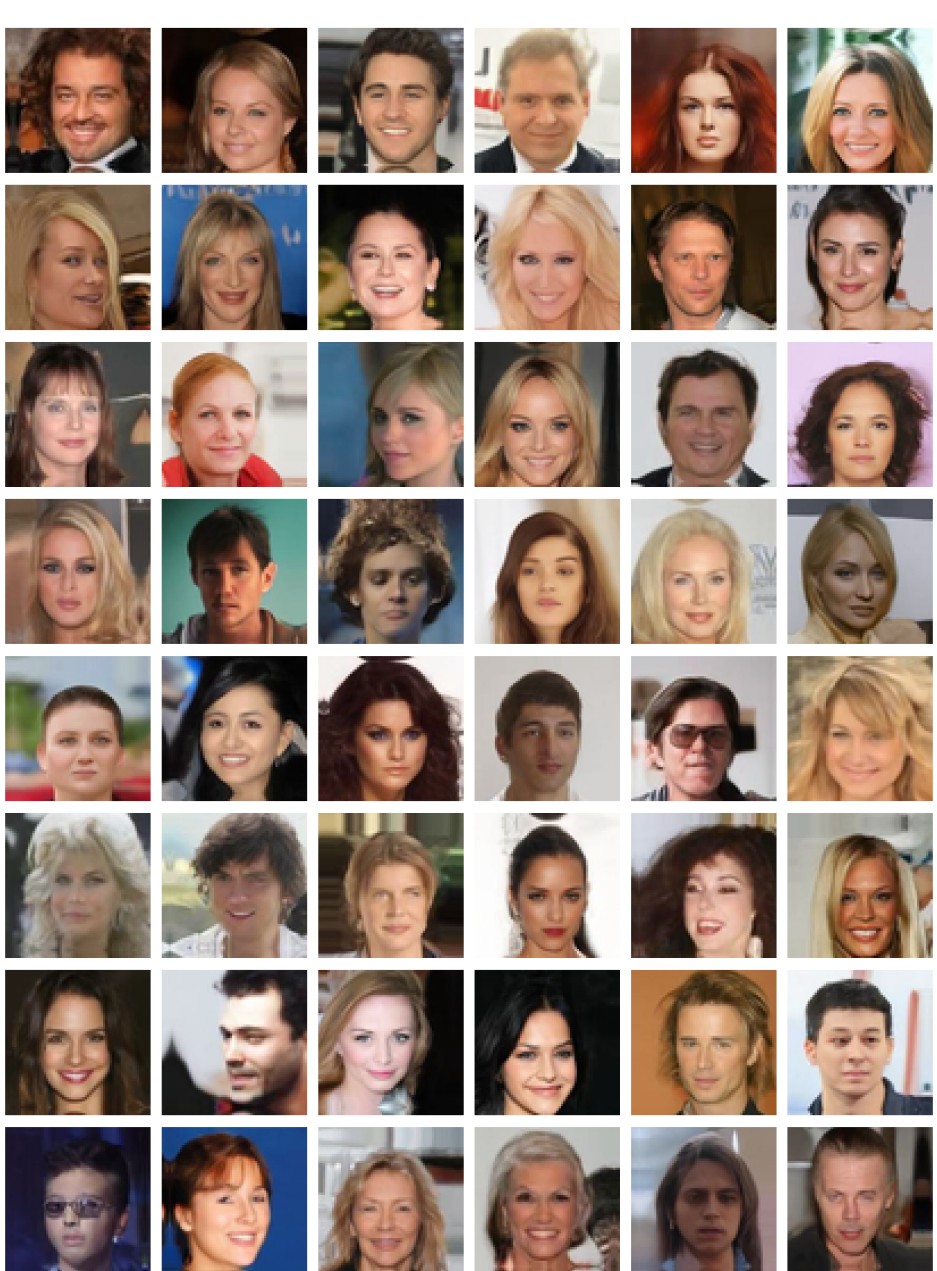

Figure 19: Unconditional samples from the experiment with CelebA dataset with $\rho = 0.5$ corruption probability.

