# OpenReview forum: "DiffEM: Learning from Corrupted Data with Diffusion Models via Expectation Maximization"
_ICLR.cc/2026/Conference — Submitted to ICLR 2026_

### Official Review · Reviewer_FK5b · 2025-10-24

**Soundness:** 3
**Presentation:** 3
**Contribution:** 2
**Rating:** 6
**Confidence:** 4

**Summary:**

The authors propose a method to train diffusion models only using corrupted/noisy observation based on expectation maximisation (EM). The idea of using EM to train diffusion models using corrupted observation is not novel
 and has been done in prior work [1],[2] (as the authors also point out in the introduction). However, previous work only trains an unconditional diffusion models and uses heuristic methods to create posterior samples for
 the E-step. In contrast, the authors make use of conditional diffusion models for the posterior sampling step.

[1] Rozet et al. "Learning Diffusion Priors from Observations by Expectation Maximization"

[2] Bai et al. "An Expectation-Maximization Algorithm for Training Clean Diffusion Models from Corrupted Observations"

**Strengths:**

- I think it is a good idea to make use of conditional diffusion models instead of heuristic approximations for posterior sampling in the E-step. In particular, in the context of non-linear forward operators $A$, where many of the heuristic methods often fail.

- The paper is well-written and easy to follow

**Weaknesses:**

- I have a few small issues with the formatting: page 2 (line 88) the subsection title "1.1 Preliminaries" is way to close to the figure subtitle, the text in Figure 1 is hard to read (small black text on a blue background)

- In your experiments you choose two degradation processes: random masking and gaussian blur. In both of these setting the heuristic conditional sampling method often perform quite good. It would be interesting to see how your approach works under non-linear
degradation operators?

- For both datasets (Cifar-10 and CelebA) you also could compare the "prior reconstruction" performance against a classical diffusion model, trained on the clean dataset. Such a comparison would be interesting to show the upper limit of performance.

**Questions:**

- In Algorithm 1 you state that you also output an unconditional model trained on the last sampled dataset $D_X^{(K-1)}$. It has been observed that training generative models on the output of another generative
  model can degrade the quality a lot, see e.g. [3], [4]. Do you also observe something similar in your experiments?
- How many samples do you need to train the unconditional model?
- Would it be possible to use classifier-free guidance to both train the conditional and unconditional model at the same time?
- EM does depend a lot on the initialisation, how do you choose the initial model?
- The noise for the Cifar-10 experiments seems to be really small ($sigma_Y^2 = 1e-6$). What happens for larger, more realistic noise?
- How does the **unconditional samples** of CelebA look like? Do you observe less diversity compared to traditional unconditional diffusion models trained on the clean CelebA data?

[3] Alemohammad et al. "Self-Consuming Generative Models Go MAD"

[4] Shumailov et al. "The Curse of Recursion: Training on Generated Data Makes Models Forget"

---

> ### Author Response · Authors · 2025-11-26
>
> We thank the Reviewer for their constructive feedback. We are glad that they appreciated the simplicity of our approach, the avoidance of heuristics, the theoretical contributions, and the broad applicability of the method. Below we address each point raised by the Reviewer.
>
> > page 2 (line 88) the subsection title "1.1 Preliminaries" is way to close to the figure subtitle
>
> This formatting issue has now been resolved.
>
> > In your experiments you choose two degradation processes: random masking and gaussian blur. In both of these setting the heuristic conditional sampling method often perform quite good. It would be interesting to see how your approach works under non-linear degradation operators?
>
> We agree this is an important point. To address it, we have added a new experiment using a non-linear degradation operator—JPEG compression. This experiment is included in Appendix Section C7. We plan to further expand this ablation with additional metrics and datasets for the camera-ready version. If the Reviewer has any preferred setups or degradation operators they would like us to evaluate, we would be happy to incorporate their suggestions.
>
> > In Algorithm 1 you state that you also output an unconditional model trained on the last sampled dataset $D_X^{(K-1)}$. It has been observed that training generative models on the output of another generative model can degrade the quality a lot, see e.g. [3], [4]. Do you also observe something similar in your experiments?
>
> Yes, we observe similar behavior: after many EM iterations, the unconditional model trained on $D_X^{(k-1)}$ begins to show gradual quality degradation, consistent with findings in [3, 4]. However, this degradation is typically very \emph{slow} and only appears after a substantial number of iterations. For CIFAR-10, running 40 EM iterations reveals a clear “sweet spot’’ with the best sample quality; artifacts only begin to appear well beyond this point (see Figure 14). These trends and representative artifacts are documented in the “MAD’’ subsection in Appendix C8. For JPEG compression with very high corruption (e.g., JPEG quality 20%), degradation occurs more quickly.
>
> > How many samples do you need to train the unconditional model?
>
> We use exactly the same number of samples as in the original corrupted dataset; no additional samples are introduced.
>
> > Would it be possible to use classifier-free guidance to both train the conditional and unconditional model at the same time?
>
> Classifier-free guidance could in principle be incorporated. However, joint training of conditional and unconditional models under CFG would require recomputing or jointly optimizing the EM targets at every iteration, which would significantly increase computational cost and complicate the EM objective. For clarity and stability, we therefore keep the two training processes separate. Investigating joint training with CFG is an interesting future direction.
>
> > The noise for the Cifar-10 experiments seems to be really small ($sigma_Y^2=1e-6$). What happens for larger, more realistic noise?
>
> The small noise level ($1\text{e}{-6}$) is used purely as a light augmentation to improve robustness, not as a realistic corruption model. For larger noise levels, the behavior is consistent: increasing $\sigma_Y$ simply modifies the corruption channel and is naturally handled by the EM updates. We have clarified this in the main text. Additionally, we added a dedicated experiment with $\rho = 0$ and $\sigma_Y = 0.2$, described in Appendix Section C10, with qualitative samples shown in Figure 17.
>
> > How does the unconditional samples of CelebA look like? Do you observe less diversity compared to traditional unconditional diffusion models trained on the clean CelebA data?
>
> We include several unconditional CelebA samples in the final section of the appendix. Qualitatively, we do not observe any noticeable loss of diversity compared to unconditional diffusion models trained on clean CelebA. To support this observation quantitatively, we added diversity and coverage metrics, under which the unconditional model remains competitive.

---

> > ### Comment · Reviewer_FK5b · 2025-11-27
> >
> > Thank you for the response and for providing the additional experiments.
> >
> > 1. MAD Subsection
> >
> > The added experiments showing FID versus EM iterations are useful. Was the unconditional model retrained from scratch on the current dataset $\mathcal{D}_X^{k}$ at each step k?
> >
> > Given that FID can degrade significantly over EM iterations (e.g., JPEG corruption in Fig. 13), do you use any form of early stopping that does not require ground-truth samples?
> >
> > How do you interpret these empirical results in light of the linear convergence claim for EM in Proposition 3?
> >
> > 2. CFG During Training
> >
> > You write:
> >
> > “However, joint training of conditional and unconditional models under CFG would require recomputing or jointly optimizing the EM targets at every iteration, which would significantly increase computational cost and complicate the EM objective.”
> >
> > I do not fully understand this argument. What I meant, was that in Algorithm 1, during the M-step, one could use the standard CFG training procedure (i.e., randomly dropping the conditioning input) to train both the conditional model (used in the next E-step) and the unconditional model simultaneously.
> > Since the unconditional model may perform poorly for a large number of EM iterations (as shown in the MAD subsection), this joint training approach could stabilize its behavior or at least avoid retraining it from scratch on $\mathcal{D}_X^{k}$?
> >
> > **Additional Questions**
> >
> > Why is “Eval 2” (Line 461) described as a reconstruction task when the comparison there concerns unconditional models?
> >
> > How exactly do you compute density, recall, precision, and coverage in your experiments?
> >
> > Table 2 (Line 404) lists “density, recall, precision, coverage, and vendi.” I assume “vendi” is an error. Also, Table 2 is not referred to in the main text at the moment.

---

> ### Author Response · Authors · 2025-11-28
>
> >Given that FID can degrade significantly over EM iterations (e.g., JPEG corruption in Fig. 13), do you use any form of early stopping that does not require ground-truth samples?
>
> For these experiments, we just always compute the FID with respect to the validation set, and report the best performance. We agree that such validation set might not exist in some settings. In the image domain, other non-distributional metrics could be used, e.g. CLIP-IQA [3] for quality. However, in the most general setting, if there is no reference set nor good evaluation metrics, it might be hard to find the optimal stopping strategy
>
> Furthermore, we have ran more experiments on JPEG compression, empirically we see that the the degradation is severe when the corruption level is very high, and for other cases it is not as much severe.
>
> >How do you interpret these empirical results in light of the linear convergence claim for EM in Proposition 3?
>
> Proposition 3 has a second additive term that captures the maximum learning error over all previous iterations of the algorithm, defined at the top of page 7 (under the Convergence rate) paragraph. As the algorithm proceeds, this term can only remain constant or increase.
>
> Empirically, we observe that running an excessive number of EM iterations indeed leads to a noticeable increase in this term. Proposition 3 explicitly places an assumption on how big this term can get – this assumption seems to be violated once we get to the “madness” regime. This makes sense – in the definition of this error term the expectation is taken over the real joint distribution. In the madness regime, the model learns to amplify its own mistakes and get far from this ideal distribution.
> This is consistent with the observations from the “diffusion models go MAD” literature. We hope this clarifies it. We added the relevant explanation to the paper to clarify this. We thank the Reviewer for raising this important point.
>
> **Using CFG in our pipeline**
>
> Apologies to the Reviewer. We previously misunderstood what the Reviewer proposed. To answer the Reviewer’s initial question, yes, CFG training (i.e., in parallel training of the conditional and the unconditional model) could be incorporated in our pipeline. Further, this would avoid the need for retraining the unconditional model from scratch.
>
> That said, we want to clarify what seems to be a misunderstanding. We do not train the unconditional model at every iteration. We only need to train the conditional model, and then at each iteration of the EM we use it to reconstruct the measurements, and then we use the reconstructed set as the training set for the next conditional model. We only perform one unconditional training at the very end (e.g., at iteration 20 of our EM pipeline) to get a final unconditional model. We agree that this part does not need to be separate and that it could be incorporated at a single step using CFG training.
>
> >The added experiments showing FID versus EM iterations are useful. Was the unconditional model retrained from scratch on the current dataset $\mathcal{D}_X^{k}$ at each step k?
>
> See explanation above. The unconditional model is indeed trained from scratch, but this only happens at the very last iteration of our training. We agree that with CFG this last training from scratch could have been avoided. For our conditional experiments, we always initialize from the model of the previous iteration.
>
> > Why is “Eval 2” (Line 461) described as a reconstruction task when the comparison there concerns unconditional models?
>
> We agree with the Reviewer that this naming is misleading. We used Reconstruction to refer to how well we capture the distribution unconditionally. This is indeed misleading. We changed this, and we thank the Reviewer for raising this point – our sincere apologies for the confusion.
>
> > How exactly do you compute density, recall, precision, and coverage in your experiments?
>
> The metrics are always with respect to the “ground-truth” distribution that we don’t have access to during training. These are metrics from the paper [2] and we use [1] for computation. We updated the paper to clarify and cite appropriately.
>
> > Table 2 (Line 404) lists “density, recall, precision, coverage, and vendi.” I assume “vendi” is an error. Also, Table 2 is not referred to in the main text at the moment.
>
> Sincere apologies for the repeated notational issues and sloppy writing – we got overwhelmed with the experiments of the Rebuttal, and we did not do a thorough check of the writing. We removed Vendi from the Table, referred to the Table from the main text and the appendix.
>
> [1] https://github.com/layer6ai-labs/dgm-eval
>
> [2] Stein et al., Exposing flaws of generative model evaluation metrics and their unfair treatment of diffusion models
>
> [3] Jianyi Wang et al., Exploring CLIP for Assessing the Look and Feel of Images

---

### Official Review · Reviewer_F3GN · 2025-11-01

**Soundness:** 2
**Presentation:** 2
**Contribution:** 2
**Rating:** 4
**Confidence:** 3

**Summary:**

This paper proposes to learning from corrupted data using the EM algorithm.

**Strengths:**

The proposed method can learn data distribution from corrupted data

**Weaknesses:**

1. Essential baselines/references not compared or discussed thoroughly. Including [1], [2]
2. The EM methods are not new, and used to train diffusion models with measurements. The novelty in this paper is just that replacing the posterior sampling with a conditional model if I am understanding it correctly.


[1] An Expectation-Maximization Algorithm for Training Clean Diffusion Models from Corrupted Observations

[2] Learning Diffusion Priors from Observations by Expectation Maximization

**Questions:**

The EM methods are not new, and used to train diffusion models with measurements. Training a conditional diffusion model usually means that it may need retraining for different measurements, and generalize not that well to OOD data. Could author discuss about this scenarios and analyze whether there is performance drop on OOD measurements or data?

---

> ### Author Response · Authors · 2025-11-26
>
> We thank the reviewer for their thoughtful comments. We address each concern below.
>
> >Essential baselines/references not compared or discussed thoroughly. Including [1], [2]
>
> Our evaluation closely follows the EM-MMPS framework [2], which we selected because it achieves superior performance to [1] and has well-documented, reproducible code. We reproduced [2] across all experiments. While we initially relied on reported results for [1] due to documentation limitations in its codebase, we have now included a direct experimental comparison on the CIFAR-10 inpainting task:
>
> | Method         |   FID  | PSNR | LPIPS |
> |----------------|--------|------|-------|
> | DiffEM (ours)  | 10.24  | 23.9 | 0.015 |
> | Method [1]     | 21.08  | 24.7 | 0.009 |
>
> It is important to note that our experimental setup is more challenging: we use a corruption probability of $\rho = 0.75$, whereas [1] uses $\rho = 0.6$. Despite this harder setting, DiffEM substantially outperforms [1] on FID while maintaining comparable PSNR and LPIPS. Additionally, we have included diversity and coverage metrics in Table 2, providing a more comprehensive evaluation than either baseline. We believe these comparisons now adequately address the concern.
>
> >The EM methods are not new, and used to train diffusion models with measurements. Training a conditional diffusion model usually means that it may need retraining for different measurements, and generalize not that well to OOD data. Could author discuss about this scenarios and analyze whether there is performance drop on OOD measurements or data?
>
> We appreciate this important critique. While we initially did not thoroughly explore out-of-distribution robustness, we have now added extensive ablations in Appendix Section C6. To evaluate OOD generalization, we fix the true corruption level at $\rho = 0.75$ and vary the model's assumed masking ratio across $\{0.65, 0.75, 0.85\}$. This setting directly tests whether the model degrades when the assumed corruption level mismatches the true data distribution.
>
> The results below demonstrate performance sensitivity to misspecification for the conditional model after 18 EM iterations:
>
> | **Assumed Masking Ratio** | **ρ = 0.65** | **ρ = 0.75** | **ρ = 0.85** |
> |---------------------------|--------------|--------------|--------------|
> | **FID**                   | 9.39         | 5.41         | 7.36         |
>
> A key finding is that the model performs best when the assumed ratio correctly specifies the true corruption level ($\rho = 0.75$ yields FID of 5.41). When misspecified, performance degrades, but the model exhibits an asymmetry: overestimating the corruption is preferable to underestimating it. This provides practical guidance for practitioners deploying the model on related (but not identical) corruption distributions. We include detailed analysis and visualizations of this behavior in the updated paper. Additional ablations on misspecification for blur corruptions will be included in the camera-ready version. We thank the reviewer for suggesting this important analysis.

---

### Official Review · Reviewer_MXuf · 2025-11-02

**Soundness:** 3
**Presentation:** 2
**Contribution:** 3
**Rating:** 4
**Confidence:** 3

**Summary:**

This paper proposes an expectation-maximization scheme for training diffusion models from so-called corrupted data. The idea is to utilize conditional diffusion models to implement the E-step as opposed to prior work (which use diffusion posterior sampling for E step). The paper provides some theoretical claims and empirical results to support these claims.

**Strengths:**

The paper provides a reasonable approach to avoid the issues related to E-step in prior art. The idea here is to use conditional diffusion models -- albeit this requires an extra training burden. I appreciate that the authors discussed this explicitly. There are some theoretical results which can be further utilized in future work for a theoretical understanding of similar schemes.

**Weaknesses:**

The paper contains many confusing parts as will be discussed in the questions. The terminology, presentation, and notation is not often very clear.

The novelty of the work is very limited: It merely replaces the E-step in prior work with something more expensive, by training dedicated samplers; I think the idea is a straightforward extension of prior work.

**Questions:**

1) Figure 1 contains a typo in the caption (reads: todo). I found this figure quite confusing at this stage of the paper. It contains lots of notation that the reader is unfamiliar with at this stage, and the figure is really not well done and clear. I would suggest getting rid of it, or replacing it with something much simpler and cleaner.

2) The mathematical setting is a bit confusing. The paper starts by talking about a prior $P_X^\star$ which is defined over latent variables. This is the standard data distribution in generative modelling setting - this is not clarified. 'Forward channel' which is easy to mistake with the 'forward process' is instead the *standard likelihood* in ML/AI papers, again suffering from obscure terminology. This is also called corrupted channel, but this is not an information theory paper. I strongly suggest you to align with standard terminology and avoid *forward channel* altogether, it is confusing.

3) After this, diffusion models are introduced, talking about some $p_0$ to be recovered. No connections are made between $p_0$ and $P_X^\star$. I think it would make sense to clarify this for the reader's benefit.

4) Section 2 details the latent variable model setting. Now what is $q_\theta(x)$? It is never introduced. What is the relationship to $p_0$ and $P_X^\star$? Is it yet another notation for the same thing (or its approximations)?

5) I think it is not clear in Section 2.2 how eq. (10) pops out in the M-step (you just say *we consider the following conditional score matching loss*). Can you give a full derivation of how E and M steps are derived in your case in Section 2.2? Does it come out of standard expectation of the joint log-likelihood? Or is it an ad-hoc loss being introduced here?

---

> ### Author Response · Authors · 2025-11-26
>
> We thank the reviewer for their detailed feedback.
>
> > The novelty of the work is very limited: It merely replaces the E-step in prior work with something more expensive, by training dedicated samplers; I think the idea is a straightforward extension of prior work.
>
> We acknowledge that our proposed algorithm is simple. That said, we view this as a virtue. Prior work relied on approximations such as the computationally expensive Moment Matching method for posterior sampling. Instead, our algorithm is exact, simple, performs better in practice and also is more compute friendly. We expect this qualities to lead to a wider adoption
>
> > Figure 1 contains a typo in the caption (reads: todo). I found this figure quite confusing at this stage of the paper. It contains lots of notation that the reader is unfamiliar with at this stage, and the figure is really not well done and clear. I would suggest getting rid of it, or replacing it with something much simpler and cleaner.
>
> We thank the reviewer for pointing this out. We sincerely apologize for the typo and have replaced Figure 1 with a more interpretable and polished version.
>
> > The mathematical setting is a bit confusing. The paper starts by talking about a prior $P_X^\star$
>  which is defined over latent variables. This is the standard data distribution in generative modelling setting - this is not clarified. 'Forward channel' which is easy to mistake with the 'forward process' is instead the *standard likelihood* in ML/AI papers, again suffering from obscure terminology. This is also called corrupted channel, but this is not an information theory paper. I strongly suggest you to align with standard terminology and avoid forward channel altogether, it is confusing.
>
> Thank you for the suggestion. We originally adopted the terminology 'prior' and 'forward channel' following Rozet et al. (2024) and the Bayesian literature. We used 'corruption channel' since in many applications the forward mechanism indeed corresponds to corruption (e.g., masking or adding Gaussian noise). If the reviewer could propose alternatives to the words “corruption channel” and “forward channel”, we would be happy to adopt the recommendation and update the text accordingly.
>
> Regarding the suggestion to use 'standard likelihood': could the reviewer clarify which references use this term? We are not aware of prior work that adopts this terminology in this context.
>
> > After this, diffusion models are introduced, talking about some $p_0$
>  to be recovered. No connections are made between $p_0$ and $P_X^\star$. I think it would make sense to clarify this for the reader's benefit.
>
> In our setup, the diffusion model is conditional and is used to learn the posterior distribution $P_{X|Y=y}^\star$. Because we model posterior rather than the prior directly, $p_0$ in the diffusion process is not intended to correspond to $P_X^\star$.  For details, please see Section 2.2.
>
> > Section 2 introduces $q_\theta(x)$ but does not explain what it is or how it relates to $p_0$ and $P_X^\star$. Is this yet another notation for the same distribution or its approximations?
>
> Following the classical EM formulation, we model the joint distribution using a parametric family $\\{q_\theta(x,y)\\}$, and it is unrelated to $p_0$. In the revision, we have explicitly defined $q_\theta$ and added a more detailed explanation of how it fits within the EM framework. Thank you for pointing that out.
>
> > I think it is not clear in Section 2.2 how eq. (10) pops out in the M-step (you just say *we consider the following conditional score matching loss*). Can you give a full derivation of how E and M steps are derived in your case in Section 2.2? Does it come out of standard expectation of the joint log-likelihood? Or is it an ad-hoc loss being introduced here?
>
> The minimizer of the loss in (10) is precisely the _conditional score function_, which is the quantity required to perform conditional sampling. We have now made this connection explicit in the revised text. This follows directly from the conditional form of Tweedie’s formula:
>
> $$s^\star(x, t \mid y) = \frac{\mathbb{E}[X_0 \mid X_t = x, Y = y] - x}{\sigma_t^2}$$
>
> which shows that the posterior score is the minimizer of the conditional score-matching objective. This identity has been used in several prior works on conditional score matching, such as SR3 [1], and is also summarized in the recent diffusion–inverse-problems survey [2].
>
> [1] "Image Super-Resolution via Iterative Refinement’’ Saharia
>
> [2] A Survey on Diffusion Models for Inverse Problems G. Daras

---

> > ### Comment · Reviewer_MXuf · 2025-11-26
> >
> > Many thanks for your response.
> >
> > > Regarding the suggestion to use 'standard likelihood': could the reviewer clarify which references use this term? We are not aware of prior work that adopts this terminology in this context.
> >
> > It must be obvious that I meant simply *likelihood* here. In line 091-092, you say
> >
> > > The forward channel (or *corruption process*) $\mathbf{Q}(\cdot | X)$ maps each point $X \in \mathcal{X}$ to a distribution over the observation space $\mathcal{Y}$.
> >
> > The distribution $\mathbf{Q}$ is simply the 'likelihood' (not 'standard likelihood') (compare it to standard way of writing it as a density $p(y|x)$) which is called as such in many papers dealing with inverse problems, I am not sure if authors really want me to list the whole Bayesian inverse problems literature. From my perspective, you simply have a conditional measure $\mathbf{Q}$ (say with density $p(y|x) = \mathcal{N}(y; A(x), \sigma_Y^2)$ to make the connection to inverse problems) which is not typically called the 'corruption process' or 'forward channel'. The former is more intuitive from a signal processing perspective. But I will stress the point to avoid using 'forward channel' in a paper using diffusion models (for which 'forward process' means an entirely different thing), this is confusing.
> >
> > > We originally adopted the terminology 'prior' and 'forward channel' following Rozet et al. (2024) and the Bayesian literature.
> >
> > I would be happy to see any central references from Bayesian inverse problems literature which uses the term 'forward channel' instead of the term likelihood.
> >
> > > Figure 1 with a more interpretable and polished version
> >
> > Many thanks for this update, I appreciate the effort. I would make the connection here clear with the training loop and how it improves the DDPM sampling by making the interaction between M step and the subsequent E step clear.
> >
> > I have some follow-up questions/suggestions
> >
> > - You define in eq. (2), (and in line 108) $Y$ as a pair of a vector and a matrix. Can you make it clear the space $Y$ lives in? ($\mathbb{R}^{d_y} \times \mathbb{R}^{d_y \times d_x}$?) Normally $d_y$ would mean the dimension of $Y$ (intuitively, it is widely used this way, and since you denote $d_x$ as the dimension of $X$. But here $Y$ is not a $d_y$-dimensional vector, is that correct?
> >
> > - In line with the comment above, please in Section 2, define the domains and ranges of $q_\theta(x, y)$ and $q_\theta(y)$.
> >
> > - In Section 2.1, you fix $A$ in $\mathbf{Q}$, but the way you define it above (in eq. (2)), this is not so (you note this is for simplicity, but only in that section?). Maybe using $\mathbf{Q}_A$ for this special case would be better.
> >
> > - In Algorithm 1, you refer to $\mathcal{D}_Y$ as 'corrupted observations' and note that you assume that the 'forward channel' (likelihood) is 'known' which I assume means $A$ is fixed. Therefore, is it correct to say you never sample $A$ throughout the training run? If so, why introduce the complicated setting in eq. (2) which certainly differs from standard way of introducing inverse problems? If you are indeed sampling $A$, where is this stage in Eq. (2)?
> >
> > Thanks in advance for your reply.

---

> ### Author Response · Authors · 2025-11-26
>
> We sincerely thank the reviewer for the prompt reply, constructive suggestions and detailed clarifications.
> **We have updated the submitted PDF to incorporate the additional feedback**.
> In the latest revision, we have made the following updates in response:
>
> - We now refer to $P_X^\star$ as the **data distribution** and $\mathbf{Q}$ as the **likelihood**, and we have removed the terms _forward channel_ and _corruption channel_ for consistency and clarity.
> - We reserve the term **prior** exclusively for $\pi^{(k)}$ in the EM iterations, to clearly distinguish it from the true data distribution $P_X^{\star}$.
> We hope these updates resolve the earlier confusion. If anything remains unclear, we are very happy to revise further. We will also update Figure 1 to better illustrate the interaction between the E-step and M-step.
>
> > You define in eq. (2), (and in line 108) $Y$ as a pair of a vector and a matrix. Can you make it clear the space $Y$ lives in? ($\mathbb{R}^{d_y} \times \mathbb{R}^{d_y \times d_x}$?) Normally $d_y$ would mean the dimension of  (intuitively, it is widely used this way, and since you denote $d_y$ as the dimension of $Y$. But here  is not a $d_y$-dimensional vector, is that correct?
> > In line with the comment above, please in Section 2, define the domains and ranges of $q_\theta(x,y)$ and $q_\theta(y)$.
>
> We thank the reviewer for this question. In Eq. (2): a clean datapoint $X$ is corrupted into $AX+\epsilon$, where $A\in \mathbb{R}^{d\times d_x}$ is the corruption matrix. Therefore, the observation $Y=(AX + \epsilon, A) \in \mathbb{R}^{d} \times \mathbb{R}^{d \times d_x}$ indeed lives in the product space $\mathbb{R}^{d} \times \mathbb{R}^{d \times d_x}$. We agree that the earlier notation $d_y$ was confusing, so we replaced it with d and explicitly clarified the space of $Y$ in Eq. (2). We have also added formal definitions of the domains of $q_\theta(x)$ and $q_\theta(x,y)$ in Section 2.
>
> > In Section 2.1, you fix $A$ in $Q_A$, but the way you define it above (in eq. (2)), this is not so (you note this is for simplicity, but only in that section?). Maybe using $Q_A$ for this special case would be better.
>
> We appreciate this suggestion. We have revised Section 2.1 to use the notation $Q_A$ consistently to denote the measurement model when $A$ is fixed, and we now explicitly state that this represents a special-case setting.
>
> > In Algorithm 1, you refer to $\mathcal{D}_Y$ as 'corrupted observations' and note that you assume that the 'forward channel' (likelihood) is 'known' which I assume means $A$ is fixed. Therefore, is it correct to say you never sample $A$ throughout the training run? If so, why introduce the complicated setting in eq. (2) which certainly differs from standard way of introducing inverse problems? If you are indeed sampling  $A$, where is this stage in Eq. (2)?
>
> We apologize for the earlier lack of clarity. In the general formulation of Eq. (2), the corruption matrix A is **randomly sampled** from a **known distribution** P_A. In Algorithm 1, the dataset $\mathcal{D}_Y$ consists of observations of the form $Y=(AX+\epsilon,A)$, where **each datapoint may have a different** A. We do not assume that $A$ is fixed; only that we have sample access to $P_A$. This is sufficient for performing the M-step, where conditional score matching requires expectations with respect to the known likelihood model.
>
>
> We deeply appreciate the reviewer's careful scrutiny and insightful questions. Your feedback has significantly improved the clarity and rigor of our presentation. We hope these revisions address the raised concerns, and we would be happy to refine the manuscript further if needed.

---

### Official Review · Reviewer_oPzx · 2025-11-04

**Soundness:** 2
**Presentation:** 3
**Contribution:** 2
**Rating:** 4
**Confidence:** 4

**Summary:**

The paper introduces DiffEM, a method for training diffusion models from corrupted observations only by learning a conditional diffusion posterior within an EM framework. Instead of treating a pre-trained unconditional prior as a fixed prior and approximating the posterior at inference, DiffEM directly trains a conditional diffusion model $q_\theta(x|y)$ with conditional score matching (M-step), using reconstructions produced by the current model (E-step). The authors give approximate monotone-improvement and convergence guarantees and demonstrate superior results over Ambient-Diffusion and EM-MMPS on masked/blurred CIFAR-10 and CelebA, alongside a computation-time analysis.

**Strengths:**

Modeling the *posterior* with a conditional diffusion directly is conceptually clean and broadly applicable to any known forward channel $Q(y|x)$, avoiding ad-hoc posterior approximations. Training reduces to standard conditional score matching; architecture/hyperparameters are transparent and close to common U-Net baselines. The paper states an identifiability assumption and proves a linear-rate EM convergence result up to learning/discretization error terms, giving useful intuition on where progress comes from. The paper decomposes cost (Tinit, K·Tft, Tu) and shows DiffEM is more time-efficient than EM-MMPS under the reported setup; warm starting from an EM-MMPS prior further improves numbers.

**Weaknesses:**

1. Many real pipelines have *model mismatch* (unknown blur kernel/noise, non-linear camera pipeline). Robustness to misspecification isn’t evaluated.
2. The M-step trains on reconstructions produced by the current model; the paper could more directly quantify diversity/coverage over EM iterations (beyond IS/FID/FD).
3. Metrics focus on distributional quality (IS/FID/FD/FDDINOv2). For cases with available ground truth (e.g., synthetic masks/blur), additional perceptual/semantic alignment analyses would strengthen the case (even if PSNR/LPIPS is cautioned against for heavy corruption).
4. The identifiability and small-error conditions underlying the convergence claim are hard to check in high-dimensional image settings; practical diagnostics would help.

**Questions:**

1. If the training/inference $Q$ differs slightly from reality (kernel width, noise variance, mask distribution), how do metrics and monotonicity behave?
2. Since theory separates learning vs. discretization error, can you report performance vs. sampling steps/schedules for the conditional model at fixed training?
3. Under equal GPU-hour budgets, how do DiffEM and EM-MMPS trade off final quality? The current tables are helpful but not strictly compute-matched.
4. Can you show JPEG or other non-linear/non-differentiable channels to underline generality?

---

> ### Author Response · Authors · 2025-11-26
>
> We thank the Reviewer for their constructive feedback. We are glad that the Reviewer appreciated the simplicity of our approach, the avoidance of heuristics, our theoretical contributions, and the broad applicability of our method. In what follows, we address some issues raised by the Reviewer.
>
> **Robustness to corruption-channel mismatch**
>
> As the Reviewer correctly points out, in many realistic settings, the forward model is not perfectly known. In such cases, it is important to evaluate the robustness of our algorithm to this mismatch. Following the Reviewer's recommendation, we conduct an ablation. Specifically, we focus on the masking corruption, we fix the dataset masking level to 0.75, and we vary the assumed masking ratio by the model. We try the following assumed masking ratios: \{0.65, 0.75, 0.85\}.
> The results are presented below (after 18 iterations of EM):
>
> | **Assumed Masking ratio**|**ρ = 0.65**|**ρ = 0.75**|**ρ = 0.85**|
> |-|-|-|-|
> | **FID**| 9.39|**5.41**| 7.36|
>
> For further analysis please refer to section C6 in paper.
>
> **Effect of misspecifying the masking ratio.**
> We fix the true corruption level at ρ = 0.75 and vary the assumed ratio ρ used by the model.
> The correct specification ρ = 0.75 yields the best result, after that ρ=0.85 and the worst one is for ρ=0.65.
>
> As shown, performance drops when there is misspecification. The important finding from this ablation is that in such cases it is better to error on the safe side, i.e. it is better to overestimate than underestimate the corruption which results in training model for a harder task. We include these results and more detailed plots in the updated version of our paper under the Section C6. More ablations on mispecificiations (e.g. for the blur experiments) will be added in the Camera Ready version of our work. We truly thank the Reviewer for suggesting these experiments, they were really helpful to the paper.
>
> > The M-step trains on reconstructions produced by the current model; the paper could more directly quantify diversity/coverage over EM iterations (beyond IS/FID/FD).
>
> We have added more evaluations and metrics in table 2, please refer to Figure 8 for plots which shows how the coverage and density evolves after each EM iteration.
>
> > Metrics focus on distributional quality (IS/FID/FD/FDDINOv2). For cases with available ground truth (e.g., synthetic masks/blur), additional perceptual/semantic alignment analyses would strengthen the case (even if PSNR/LPIPS is cautioned against for heavy corruption).
>
> Yes, the reviewer is correct indeed we are interested in learning the distribution and not looking fore reconstruction models. The goal of the paper is to learn the underlying latent distribution, thus we chose to mostly focus on distributional metrics as done in the [1]. Currently we added more metrics that is available in table 2, but if the reviewer is particularly interested in PSNR and LPIPS comparing to [2] we have better results as shown in the table:
>
> | Method|FID|PSNR|LPIPS|
> |-|-|-|-|
> | DiffEM (ours) | 10.24  | 23.9 | 0.015 |
> | Theirs|21.08|24.7 | 0.009 |
>
> It is important to note that our experimental setup is more challenging: we use a corruption probability of ρ=0.75, whereas [1] uses ρ=0.6. Despite this harder setting, DiffEM substantially outperforms [1] on FID while maintaining comparable PSNR and LPIPS.
>
> **Comparing Under Same GPU Hour Compute**
>
> The reviewer suggest we include we fix different GPU hours and for each evaluate the end result of our algorithm and compare it to other work. We have added Figure 2 which illustrates the comparison between DiffEM and EM-MMPS. The plot shows that our approach is much more compute friendly. We thank the reviewer for pointing this out.
>
> > Since theory separates learning vs. discretization error, can you report performance vs. sampling steps/schedules for the conditional model at fixed training?
>
> We agree this is an important analysis. We now vary the sampler step count over {64, 128, 256, 512} and, for each choice, evaluate the trained conditional model across EM iterations. The corresponding plot has been added in figure 15. It confirms the theoretical separation between discretization and learning errors: performance improves systematically with more sampling steps while preserving the EM monotonicity structure.
>
> > Can you show JPEG or other non-linear/non-differentiable channels to underline generality?
> We have added an additional experiment using a JPEG corruption operator to demonstrate generality to non-linear and non-differentiable channels. The setup and results are included in the appendix section C7.
>
> [1] Rozet et al. "Learning Diffusion Priors from Observations by Expectation Maximization"
>
> [2] Daras et al. "Ambient Diffusion: Learning Clean Distributions from Corrupted Data"

---

### Meta-Review · Area_Chair_nrqU · 2026-01-01

**Summary:**

The submission proposes an EM-style procedure for learning diffusion models from corrupted observations, with the key design choice of implementing the E-step via an additional conditional diffusion model rather than posterior sampling as in prior EM-based diffusion training.

Reviewers acknowledged that the approach is potentially useful and appreciated the inclusion of theoretical insights and empirical evidence; the authors also made several revisions that improved clarity (e.g., fixing an early figure issue, tightening definitions/notation, and aligning terminology with standard likelihood-based language).

However, the overall assessment remains marginally below the acceptance threshold because the central methodological contribution is viewed as a relatively incremental (and potentially computationally more expensive) substitution for the E-step in existing EM diffusion frameworks, with limited novelty beyond prior art; moreover, substantial presentation and notation confusion persisted throughout the review, weakening accessibility and confidence in the exposition.

Given the combination of limited novelty and clarity issues, I recommend rejection at this time despite positive aspects and responsive revisions.

**Reviewer Concerns:**

* Limited novelty / incremental contribution: The main idea largely replaces the E-step used in prior EM–diffusion methods with a conditional diffusion model (often more expensive due to extra training), which reads as a straightforward extension rather than a new conceptual advance.


* Unclear/confusing exposition: Terminology, notation, and overall presentation were frequently confusing, making the method hard to follow and reducing confidence in the claims.

The authors partially addressed this issue in the revision.

**Reviewer Scores:**

none would raise the score

---

### Decision · Program_Chairs · 2026-01-26

Reject